# SCALING LAWS FOR PRECISION

**Tanishq Kumar**[*1]   **Zachary Ankner**[* 3,4]   **Benjamin F. Spector**[2]   **Blake Bordelon**[1]
**Niklas Muennighoff**[2]   **Mansheej Paul**[4]   **Cengiz Pehlevan**[1]   **Christopher Ré**[2]
**Aditi Raghunathan**[5]

[1]Harvard University   [2]Stanford University
[3]MIT   [4]Databricks   [5]Carnegie Mellon University

## ABSTRACT

Low precision training and inference affect both the quality and cost of language models, but current scaling laws do not account for this. In this work, we devise "precision-aware" scaling laws for both training and inference. We propose that training in lower precision reduces the model's *effective parameter count*, allowing us to predict the additional loss incurred from training in low precision and post-train quantization. For inference, we find that the degradation introduced by post-training quantization increases as models are trained on more data, eventually making additional pretraining data actively harmful. For training, our scaling laws allow us to predict the loss of a model with different parts in different precisions, and suggest that training *larger* models in *lower* precision may be compute optimal. We unify the scaling laws for post and pretraining quantization to arrive at a single functional form that predicts degradation from training and inference in varied precisions. We fit on over 465 pretraining runs and validate our predictions on model sizes up to 1.7B parameters trained on up to 26B tokens.

## 1 INTRODUCTION

Scale has emerged as a central driver of progress in deep learning (Brown et al., 2020). Key work on scaling (Kaplan et al., 2020; Hoffmann et al., 2022) studied tradeoffs between model/dataset size to balance performance and compute. However, the precision in which models are trained and served is an important third factor that contributes to both cost and performance. Deep learning is trending towards lower precision: current frontier models like the Llama-3 series are trained in BF16 (Dubey et al., 2024), and there is widespread effort to move the pretraining paradigm to FP8 (Micikevicius et al., 2022). The next generation of hardware will support FP4, and advances in weight-only quantization have led to training in binary and ternary at scale (Ma et al., 2024; Wang et al., 2023). How far will these paradigms go? Specifically, we ask:

> *What are the tradeoffs between precision, parameters, and data?*
> *How do they compare for pretraining and inference?*

Studying scaling in precision is challenging because work on scaling laws generally aims to drop fine-grained implementation details in pursuit of universal functional forms while work on quantization generally does the opposite, focuses on the details: how quantization is done, with what type, to what part of the model. In seeking a balance, we consider a variety of plausible functional forms, and choose one that abstracts implementation details of quantization away from loss scaling, allowing us to predict loss scaling in many situations of practical interest. This functional form that posits bit precision and parameter count interchangeably contribute to a model's "effective parameter count," $N_{\text{eff}}$, and implementation details like which parts of a model are quantized to what precision, interact with loss scaling only through their effect on this quantity.

Overall, we study the scaling of the effects of precision on loss as we vary data and parameters, both *during* and *after* training. We first study how the degradation induced by post-train quantization scales with parameters and data. We find that the degradation increases with data, so that for

---

*Equal contribution. Correspondence to `tkumar@college.harvard.edu`.

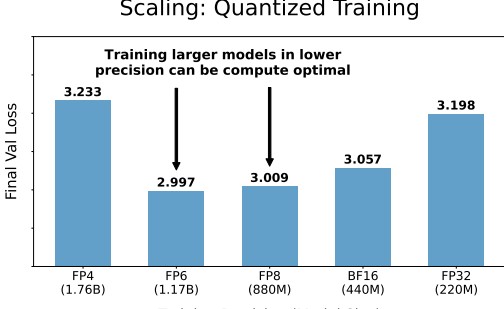

Figure 1: Schematic of key findings. (Left) Training a fixed model size to various data budgets in BF16 and quantizing weights at the end. We find that degradation due to post-train quantization increases with tokens seen during pretraining, so that eventually **additional pretraining data can be harmful**. (Right) Our scaling suggests **training *larger* models in *lower* precision can be compute-optimal** according to the cost model in Section 4.3. Weights, activations, attention quantized, all models trained on the same data budget, details in Appendix J.

a fixed model, training on additional data after a certain point can be actively harmful if the model will be quantized after training. We then shift our focus to quantized training, examining both the quantization-aware-training (weights only) and low-precision training (weights, activations, attention all quantized) settings. Our scaling laws for pretraining suggest that the compute-optimal pretraining precision is in general independent of compute budget. Surprisingly, however, this independence ceases to be true if model size is constrained, in which case the compute-optimal precision grows slowly in compute.

In all, we pretrain a suite of 465 language models in 3 to 16 bit precisions, as well as post-train quantize each to multiple precisions. For a language model with $N$ parameters, trained on $D$ tokens with training precision $P_{\text{train}}$, and post-train weight precision $P_{\text{post}}$, we ultimately find a unified scaling law that takes the following form:

$$L(N, D, P_{\text{train}}, P_{\text{post}}) = \underbrace{\underbrace{AN_{\text{eff}}^{-\alpha}}_{\text{Training-time Effects}} + BD^{-\beta} + E}_{\text{Usual Chinchilla form}} + \underbrace{\delta_{\text{PTQ}}(N_{\text{eff}}, D, P_{\text{train}}, P_{\text{post}})}_{\text{Post-Training Effects}} \quad (1)$$

where $A, B, E, \alpha, \beta$ are positive fitted constants, and $\delta_{\text{PTQ}}$ refers to the loss degradation induced by post-training quantization before inference. Altogether, our results for post-train quantization illustrate how **more pretraining FLOPs do not always lead to better models** at inference-time, and our results for low-precision pretraining suggest **that both the standard practice of training models in 16-bit, and the race to extremely low (sub 4-bit) pretraining precision, may be suboptimal.**

## 2 Background, Related Work, and Setup

**Notation.** Throughout, $D$ denotes dataset size in tokens and $N$ denotes model size in parameters. $P_{\text{w}}, P_{\text{a}}, P_{\text{kv}}$ refer to the bit precision, in integer-type, of the weights, activations, and key-value cache ("attention")[1] during *training*, and $P_{\text{post}}$ refers to the precision we post-train quantize (PTQ) weights to at the end for model inference. When $P$ or $P_{\text{train}}$ is used without reference to a part of the model, all three model parts are tied to the same precision. The inference-time loss degradation induced by post-train quantization will be denoted $\delta_{\text{PTQ}}(N, D, P_{\text{train}}, P_{\text{post}})$, and it is defined as the change in loss from performing post-training quantization compared to the end of pretraining. We use "high precision" to mean 16-bit or above.

---

[1] We study KV, rather than QKV, because understanding scaling in the KV cache alone is important for many inference settings. For pretraining claims in Section 4.3, we quantize the entire attention computation, including queries, finding additionally quantizing the query vectors makes a negligible difference to scaling.

## 2.1 Quantization Fundamentals: How, What, When

**The Problem: Compute vs Memory-Bound Workloads.** Most deep learning workloads are bottlenecked by either *compute*, in the form of matrix multiplications, or *memory bandwidth*, in the form of data movement between different parts of the GPU. Different types of workloads have different bottlenecks: most time is spent doing large matrix multiplications during pretraining, so it is compute-bound; in contrast, small-batch inference is bandwidth-bound by model weights; long-sequence decoding is bandwidth-bound by KV cache, etc. **This motivates studying scaling in the training precision of the (weights, activations, KV cache) both in isolation and in combination.**

**Quantization: How.** Quantization of an operation typically refers to rounding of values in matrices involved in some computation on the forward or backward pass, depending on what is quantized, and when. Quantization is usually done to integer or floating-point type.

**Quantization: What.** *Only weights. "Quantization-aware training"* Quantizing only weights during training does not offer any compute savings because matrix multiplications are still done in high precision. However, this is commonly done to allow weights to adapt to low precision so they can be served at very low precision at inference-time, thereby alleviating memory bottlenecks (Ma et al., 2024; Wang et al., 2023). We will refer to this as "quantization-aware-training" and defer additional discussion to Appendix D.

*Weights, activations, attention. "Low-precision training"* Quantizing and activations and attention in addition to weights allows for compute gains because matrix multiplications can be done in low precision (if the hardware supports it) since everything is in the same precision. We will refer to this setting as "low-precision training" to distinguish it from quantization-aware training.

**Quantization: When.** Quantization can be done *during* or *after* training. In practice, when seeking to reduce inference-time memory costs, one first attempts post-train quantization. If that degrades the model too much, quantization-aware-training is used. Post-train quantization is typically only applied to model weights (Frantar et al., 2022; Dettmers et al., 2022; Lin et al., 2023; Xiao et al., 2023). To reduce pretraining costs, low-precision-training is needed. We will study scaling laws for post-training quantization in Section 3, for quantized training in Section 4 (examining both quantization-aware training and low precision training) and unify the two in Section 5. The numerical values of all our fitted constants can be found in Appendix K.

## 2.2 Scaling Laws and Parametric Fits

**Scaling Laws.** Hoffmann et al. (2022) model loss scaling using the functional form $L(N, D) = AN^{-\alpha} + BD^{-\beta} + E$ where $A, B, \alpha, \beta, E$ are positive fitted constants, finding that data and parameters should be scaled in roughly equal proportion as more compute becomes available. We will refer to the scaling of (Hoffmann et al., 2022) as "Chinchilla-optimal" or just "Chinchilla" and note this is often used colloquially as $D/N \approx 20$ being pretraining compute-optimal. On the theoretical front, work on scaling laws (Bahri et al., 2024; Bordelon et al., 2024; Lin et al., 2024b) finds that noise to various parts of model or data affects loss in a predictable way. While previous works have explored the scaling behavior of post-training quantization in terms of total model bits (Dettmers & Zettlemoyer, 2023) and knowledge capacity (Allen-Zhu & Li, 2024), we focus instead on data scaling. We note that in general the exact fitted values of all coefficients and exponents can vary drastically based on small implementation differences: Besiroglu et al. (2024) find different constants when attempting to replicate (Hoffmann et al., 2022), Sardana & Frankle (2023) fit coefficients $A, B$ of different orders of magnitude. For this reason, we emphasize our contribution is not the numerical values we fit, but the trends and functional forms we identify.

**Overtraining.** In practice, accounting for inference costs means training smaller models for substantially longer than Chinchilla-optimal (Sardana & Frankle, 2023; Gadre et al., 2024). For instance, Llama-3-8B is trained to $D/N \approx 2000$ (Dubey et al., 2024) and the Gemma-2 series up to $D/N > 1000$ (Team et al., 2024). We refer to such models as "overtrained" in this paper, with the token/parameter ratio $D/N$ being a key quantity throughout. Work on inference-time compute (Snell et al., 2024; Brown et al., 2024) and on synthetic and multimodal data (Yang et al., 2024; Fan et al., 2024; Bauer et al., 2024) suggests future models may be even more overtrained. Therefore, modern work on scale must consider ratios much larger than Chinchilla-optimal, and in this work

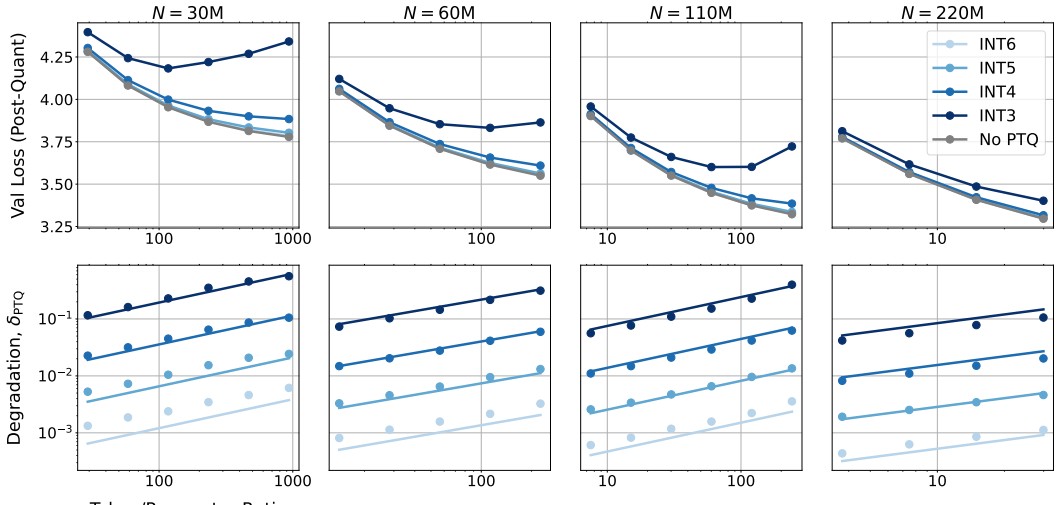

Figure 2: Loss degradation from PTQ increases with data. Top row is loss after PTQ, bottom row is loss degradation compared to end of training, before PTQ. The top row is thus the gray line in each plot plus the corresponding value in the bottom row. We can see that degradation grows with data, bottom row is fitted with Equation 2. For $D/N$ sufficiently large (left), loss can increase in data. Even at lower $D/N$, where post-quant loss continues to decrease with data, the value of data is reduced compare to the baseline. $R^2 = 0.97$ over all fitted points (bottom row).

we perform experiments up to $D/N \approx 10^3$ and analyze the predictions found by our scaling law for up to $D/N \approx 10^5$. See Appendix B for additional related work.

## 2.3 SETUP

We train and evaluate a suite of OLMo-style models on the Dolma V1.7 dataset (Groeneveld et al., 2024; Soldaini et al., 2024), using a standard Transformer++ implementation; see Appendix A for hyperparameters and ablations. Our experiments consist of a sweep of language model pretraining runs over $N \in [30, 60, 110, 220]$ million parameters (non-embedding) and $D \in [1.5, 3, 6, 13, 26]$ billion tokens. Our model sizes are relatively small because we train up to a very high $D/N \approx 10^3$ to study data scaling and set off over 20 runs at every $(N, D)$: we sweep 8 values of precision for each of the (weights, activations, attention).

## 3 SCALING LAWS FOR POST-TRAIN QUANTIZATION

The easiest and most common quantization technique is post-train quantizing a model off-the-shelf (Chee et al., 2024; Huang et al., 2024; Dettmers et al., 2022; Lin et al., 2023; Xiao et al., 2023). In this section, we consider models trained in BF16 and use GPTQ (Frantar et al., 2022) to post-train quantize them, replicating our findings with two other methods in Appendix F. We quantify the resulting loss degradation $\delta_{\text{PTQ}}$, finding that post-train quantization scales poorly in data.

### 3.1 OVERTRAINED MODELS DEGRADE MORE WHEN POST-TRAIN QUANTIZED

We consider different model sizes (columns) trained on various data budgets (x-axis of each plot) and plot in Figure 2 both the loss after post-train quantization (top row) and the degradation incurred relative to end of training (bottom row). We find that the degradation $\delta_{\text{PTQ}}$ increases in training data size across all model sizes, but that for a fixed dataset size larger models incur a smaller degradation. We additionally observe that $\delta_{\text{PTQ}}$ increases exponentially as we decrease the precision we quantize to. Based on these observations we model $\delta_{\text{PTQ}}$ as taking the form:

$$\delta_{\text{PTQ}}(N, D, P_{\text{post}}) = C_T \left( \frac{D^{\gamma_D}}{N^{\gamma_N}} \right) e^{-P_{\text{post}}/\gamma_{\text{post}}} \qquad (2)$$

where $C_T, \gamma_D, \gamma_N, \gamma_{\text{post}}$ are positive fitted constants. As we find the fitted values of $\gamma_D$ and $\gamma_N$ to be similar (see Appendix K for numerical values), we can think of this as an approximate power law in the token/parameter ratio $D/N$. The intuition for this poor data scaling might be that as models train on more data, they compress more information into their weights, so that perturbations to weights in the form of quantization are more harmful to loss, all else equal. We discuss formal theoretical interpretations in Appendix H.

This finding implies that for models that will be post-train quantized, *there exists an amount of pretraining data beyond which additional data is actively harmful to performance at inference-time* (see top-left, Figure 2). This can be defined as the point where additional data increases post-train degradation more than it decreases loss during pretraining. We solve analytically for this critical data size in Appendix E, as well analyze a cost model for workloads where inference-cost is the primary concern. We thus summarize our first scaling finding as follows.

> **Finding 1.** Overtrained language models are more sensitive to post-training quantization. For models trained in BF16 or above, we can model this loss degradation as
>
> $$\delta_{\text{PTQ}}(N, D, P_{\text{post}}) = C_T \left( \frac{D^{\gamma_D}}{N^{\gamma_N}} \right) e^{-P_{\text{post}}/\gamma_{\text{post}}}$$
>
> where $C_T, \gamma_D, \gamma_N, \gamma_{\text{post}}$ are positive fitted constants. This implies that when $D/N$ is sufficiently large, or $P_{\text{post}}$ sufficiently small, loss after quantization can increase as models are pretrained for longer, as in Figure 2. We will revisit and modify Equation 2 in Section 5 to account for the effects of *training* in low-precision on $\delta_{\text{PTQ}}$.

## 4 SCALING LAWS FOR QUANTIZED TRAINING

In this section we study pretraining with weights, activations, and KV cache in various precisions. Importantly, only training precision, not test-time precision, is varied in this section; we discuss the interaction between train and test-time precision in Section 5. We sweep the training precisions of the weights, activations, and KV cache $P_{\text{w}}, P_{\text{a}}, P_{\text{kv}} \in [3, 12]$ individually, as well as training BF16 baselines. We also pretrain models with arbitrary combinations of $P_{\text{w}}, P_{\text{a}}, P_{\text{kv}}$ to validate our scaling laws. To perform quantization during training, we quantize the forward pass in integer type unless otherwise noted, see Appendix D for implementation details.

### 4.1 QUANTIZATION-AWARE-TRAINING: QUANTIZING WEIGHTS DURING TRAINING HAS A CONSISTENT AND PREDICTABLE EFFECT

We first examine the trade-off between weight precision $P_{\text{w}}$ and parameters $N$ while holding $P_{\text{a}} = P_{\text{kv}}$ fixed at high precision. We fix $D = 13\text{B}$ tokens and perform a grid sweep over combinations of $N$ and $P_{\text{w}}$. We plot the resulting IsoLoss contours where we linearly interpolate the final loss values in Figure 3. We observe that the bit precision of the weights can be traded off for the number of parameters, i.e., a model with smaller $N$ but larger $P_{\text{w}}$ can achieve the same loss as a model with larger $N$ but smaller $P_{\text{w}}$. Additionally, we find that the gains from increasing the bit precision of the weights are large at lower precisions but saturate at higher precisions (typically around 6-7 bits per weight).

In line with the empirical trends in Figure 3, we find the best fit for the tradeoff between weight precision and parameters is $N_{\text{eff}}(N, P_{\text{w}}) = N(1 - e^{-P_{\text{w}}/\gamma_{\text{w}}})$, where $\gamma_{\text{w}}$ is a fitted constant measuring the sensitivity of model weights (alternative fits explored in Appendix K). We therefore modify Chinchilla scaling to account for $N_{\text{eff}}$ by making the substitution $N \mapsto N_{\text{eff}}(N, P_{\text{w}})$, giving the modified form:

$$L(N, D) = A[N(1 - e^{-P_{\text{w}}/\gamma_{\text{w}}})]^{-\alpha} + BD^{-\beta} + E \tag{3}$$

where we recall that $A, B, E, \alpha, \beta$ are fitted positive constants in the usual Chinchilla scaling form, and $\gamma_{\text{w}}$ is a fitted constant we introduce. We plot the predictions of our fit compared to observed values in Figure 4 for a range of $(N, D)$.

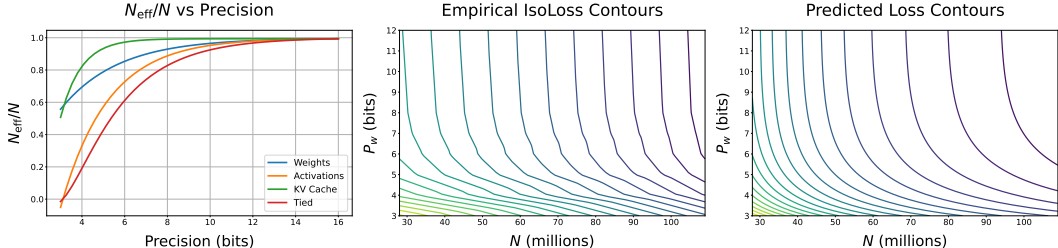

Figure 3: (Left) $N_{\text{eff}}/N$ from our final scaling law. Our fit of $N_{\text{eff}}(N, P_{\text{w}})$ in this section is the first step towards this (blue). Empirical (center) and predicted (right) IsoLoss contours illustrating the precision-parameter tradeoff. Y-axis is weight precision during quantized training. All runs plotted trained on $D = 13\text{B}$ tokens. Predictions from a fitted version of Equation 3, darker lines correspond to lower loss.

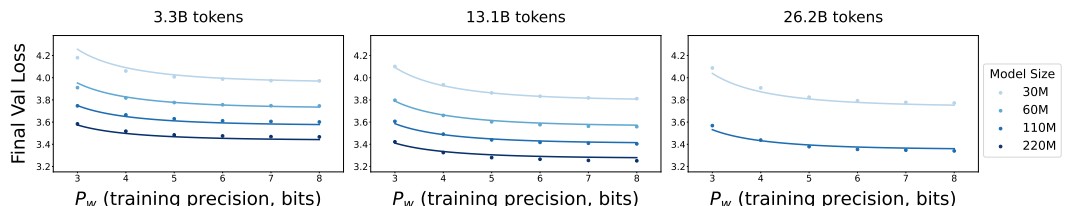

Figure 4: Predicting final validation losses $L(N, D, P_{\text{w}})$ for various $N, D, P_{\text{w}}$ to test our proposed functional form. Points are experimental values, lines are predictions of a single parametric fit of the form in Equation 3. We train only two model sizes at 26B due to compute constraints.

## 4.2 Low-Precision-Training: The Effects of Quantizing Weights, Activations, and Attention are Compositional and Multiplicative

Quantization-aware training does not change the cost of pretraining. This is because modern GPUs require inputs to a matrix multiplication to have the same precision, i.e. $P_{\text{w}} = P_{\text{a}} = P_{\text{kv}}$ (Micikevicius et al., 2022). To understand the interplay between precision and pretraining compute we must now analyze the scaling behavior of $P_{\text{a}}$ and $P_{\text{kv}}$ as well. Note that in our training experiments, we only quantize on the forward pass to ensure a fair comparison between quantization-aware-training (weights only) and the additional quantization to activations/KV cache, see Appendix D.

**Precision of activations and KV cache affect loss in a similar way.** We first verify in Appendix Figure 20 that varying $P_{\text{a}}$ and $P_{\text{kv}}$ in isolation give rise to scaling behavior that is best fit by a functional form analogous to the form for $P_{\text{w}}$ (Equation 3, Figure 5, left).

We refer to the scaling coefficients computed by varying the precision of just one part of the model at a time as *marginally fitted constants*, and those found by fitting on runs that include multiple model components in low precision at the same time as *jointly fitted constants*.

**Constants fitted marginally and jointly make similarly good predictions**. We now turn our attention to understanding the interactions between weights, activations, and attention. If the effects of quantizing weights, activations, and attention are independent, then a factorized, multiplicative interaction of the following form is a natural proposal.

$$N_{\text{eff}}(P) = N\left(1 - e^{-P_{\text{w}}/\gamma_{\text{w}}}\right)\left(1 - e^{-P_{\text{a}}/\gamma_{\text{a}}}\right)\left(1 - e^{-P_{\text{kv}}/\gamma_{\text{kv}}}\right) \tag{4}$$

We test whether this independence approximately holds by comparing the predictive power of a model with marginally fitted constants and a model with jointly fitted constants. We show the predictive power of both models in Figure 5(b, c), finding that both methods for fitting constants have approximately the same predictive power. These results suggest that the independence assumption is reasonable. We both present further evidence that this "factorized" functional form is a strong fit to the data as well as discuss alternative factorization schemes in Appendix M.

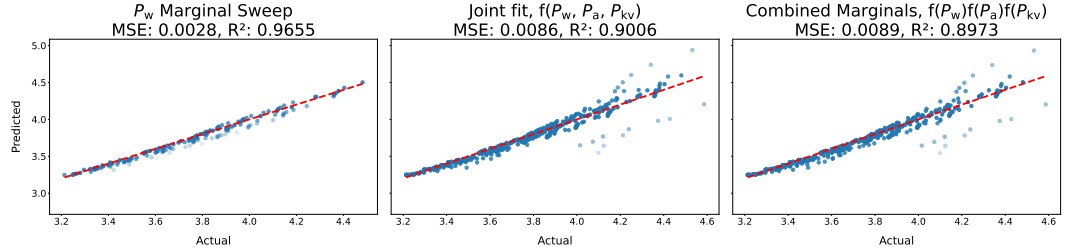

Figure 5: (Left) Predicted loss based on fitted values with Equation 4. (center) Fitting $\gamma$ parameters jointly on sweeps with combinations of precisions vs (right) fitting them on "marginal" sweeps where only one model part is in low precision at a time. Outliers are those at extremely low precision whose training runs are sometimes unstable.

---

**Finding 2.** The effects of quantizing the weights, activations, and KV cache during training are well modeled as independent and multiplicative so that

$$L(N, D, P_{\mathrm{w}}, P_{\mathrm{a}}, P_{\mathrm{kv}}) = AN_{\mathrm{eff}}^{-\alpha} + BD^{-\beta} + E$$

where

$$N_{\mathrm{eff}}(P_{\mathrm{w}}, P_{\mathrm{a}}, P_{\mathrm{kv}}) = N(1 - e^{-P_{\mathrm{w}}/\gamma_{\mathrm{w}}})(1 - e^{-P_{\mathrm{a}}/\gamma_{\mathrm{a}}})(1 - e^{-P_{\mathrm{kv}}/\gamma_{\mathrm{kv}}})$$

for which we fit constants $\gamma_{\mathrm{w}}, \gamma_{\mathrm{a}}, \gamma_{\mathrm{kv}}$ that reflect the different sensitivities of weights, activations, and KV cache. If the three precisions are set to the same value $P$, as in pretraining, this simplifies to $N_{\mathrm{eff}}(P) \approx N(1 - e^{-P/\bar{\gamma}})^3$ where $\bar{\gamma}$ is the average of the three parameters. We visualize this functional form with our fitted values in Figure 3 (left).

---

## 4.3 Implications For Pretraining

When training in a precision $P$, meaning $P_{\mathrm{w}} = P_{\mathrm{a}} = P_{\mathrm{kv}} = P$, compute cost scales linearly in $P$ (Abdelkhalik et al., 2022)[2]. Hoffmann et al. (2022) performed all experiments in 16-bit precision and use a cost model of $C = 6ND$ FLOPs. We generalize this to $C = \frac{6}{16}NDP$ to account for the linear relation between compute and precision, which reduces to the Chinchilla cost function for $P = 16$. We now examine three practically relevant variants of the following optimization problem.

$$\min_{N, D, P} L(N, D, P) = A[N(1 - e^{-P/\gamma})^3]^{-\alpha} + BD^{-\beta} + E \text{ subject to } C = \frac{6}{16}NDP \quad (5)$$

Since derivations are algebraically involved, we will work up to proportionality and verify proposed solutions numerically. See Appendix E for mathematical details. We note that the implications of our functional form are true no matter the scale at which future experiments are done, but the numerical values we predict depend on our fitted constants which are fitted on smaller-scale, integer-type experiments.

### 4.3.1 If You Must Train In Low Precision, Increase Parameters Before Data

**Minimizing** $L(N, D)$ **with** $P$ **fixed, subject to** $C \propto NDP$. We get with some algebra that at precision $P$ and compute budget $C$, the optimal allocations $N^*, D^*$ of parameters and data relative to Chinchilla-optimal $N_{\mathrm{Ch}}, D_{\mathrm{Ch}}$ will be given by

$$\frac{N^*(P, C)}{N_{\mathrm{Ch}}(C)} \propto \left[1 - e^{-P/\bar{\gamma}}\right]^{-\frac{3\alpha}{\alpha + \beta}} P^{-\frac{\beta}{\alpha + \beta}} \text{ and } \frac{D^*(P, C)}{D_{\mathrm{Ch}}(C)} \propto \left[1 - e^{-P/\bar{\gamma}}\right]^{\frac{3\alpha}{\alpha + \beta}} P^{\frac{\beta}{\alpha + \beta}} \quad (6)$$

**which suggests as precision of training decreases at fixed compute, we should increase parameters and decrease data**. The interpretation of this is that at very low precisions, our effective parameter count vanishes so that increasing parameter count is compute-optimal since data egregiously outstrips effective parameters.

---

[2]In practice, the gains are less than linear due to systems overhead.

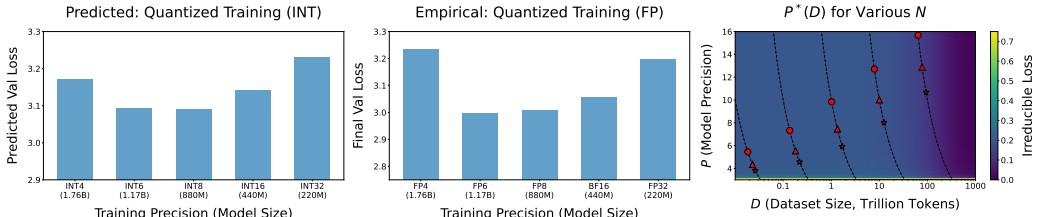

Figure 6: Scaling law predictions (left, fitted on integer type) vs empirical values (right, floating-point type). Precision of weights, activations, attention fixed to $P_{\text{train}}$. Predictions closely match the empirical trend, but are shifted up by a small amount since floating-point is a more expressive type and will incur lower loss at the same precision. (Right) When $N$ is held fixed, compute-optimal precision increases approximately logarithmically with data. Markers correspond to predicted compute-optimal precision for Llama-3 (8b, 70b, 405b), denoted by (circle, triangle, star) at each IsoFLOP (lines), illustrating how compute-optimal precision increases in data when model size is held fixed.

### 4.3.2 COMPUTE-OPTIMAL PRETRAINING PRECISION IS IN GENERAL INDEPENDENT OF COMPUTE

**Jointly minimizing** $L(N, D, P)$ **with** $C \propto NDP$. This is the setting of pretraining without constraints on $N, D, P$ except for a fixed compute budget. Solving this joint minimization problem gives an implicit equation for $P^*(C)$. Denoting $u(P) = [1 - e^{-P/\bar{\gamma}}]^{-3\alpha}$, we find (see Appendix E) that this equation takes the form

$$\frac{3\alpha}{\bar{\gamma}} \, u(P)^{\frac{3\alpha+1}{3\alpha}} e^{-P/\bar{\gamma}} = P^{-1} u(P) \tag{7}$$

which reveals that in general the optimal pretraining precision is independent of compute budget. This suggests that compute-optimal precision should be held fixed to $P^*$ while $N, D$ are scaled according to Equation 6. We find this $P^*$ to be around 7-8 bits when fitting our scaling law on runs with quantization done to integer type. This has two consequences: first, this means **the de-facto practice of training models in 16-bit may be suboptimal.** Second, **the race to low-precision training may have to stop before going below 4-bits**, since this would force model sizes to become disproportionately (more than 4x) larger to maintain loss scaling (see Figure 3, left).

We test our predictions in Figure 6 at a larger scale. We train compute-matched models at various parameter count and precision ranging from FP4 to FP32 and 220M to 1.6B parameters. We train in floating-point type since that is standard in pretraining (Groeneveld et al., 2024; Deitke et al., 2024), though our scaling laws are fitted on integer type. We plot our predicted trend in Figure 6 (left) and the empirical values in the middle. We find that scaling fits on integer type are a strong fit until 4-bit precision, at which points the difference between the two types becomes more apparent. The matching of qualitative trends throughout, with the optimum being close to the predicted optimum of $P^*$ near 7-8 bits suggests that similar scaling laws may exist across types. We initiate a similar analysis for floating-point type in Appendix **??**.

### 4.3.3 BUT COMPUTE-OPTIMAL PRETRAINING PRECISION CAN INCREASE IN COMPUTE IF MODEL SIZE $N$ IS CONSTRAINED

**Minimizing** $L(D, P)$ **with** $N$ **fixed, subject to** $C \propto NDP$. A common use case in practice is to train a suite of models of various sizes on similar data. The Llama-3 and Gemma-2 series (Dubey et al., 2024; Team et al., 2024) are examples. In this setting, $N$ is fixed in advance and only $D, P$ are jointly optimized. Surprisingly, our scaling laws predict that models of differing sizes should *not* necessarily be trained in the same precision, and that compute-optimal precision scales as $P^*(C) \propto \log C$. Since $N$ is held constant and we show in Appendix E that $\log C \approx \log D$ in proportion, we can write $P^*(C) \propto \log(D/N)$. The intuition for this is that, for a fixed $N$, precision acts as a new lever to bring highly overtrained models closer to pretraining optimality[3] by reducing $D/N_{\text{eff}}$.

---

[3]An important subtlety here is that since models are overtrained for inference, we want to keep the cost of a forward pass—which is proportional to $NP$—fixed, not just $N$. While $NP$ is the same for both a model of $N_0$

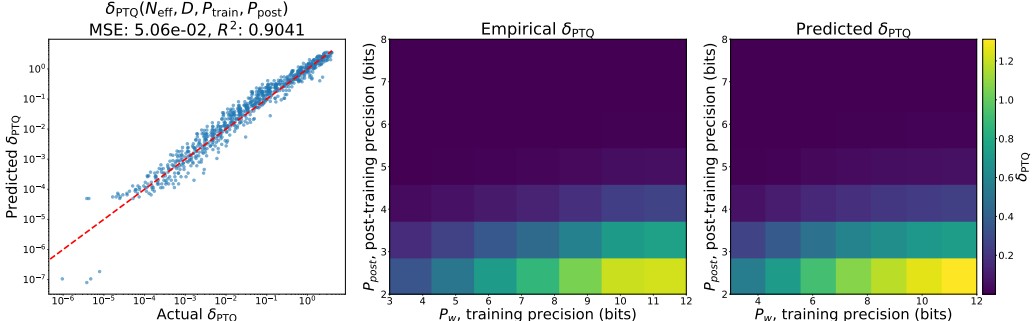

Figure 7: Combined plots for predicting degradation. (Left) demonstrates the quality of our fit on all our runs, including all combinations of pre and post-training precisions. (Center, right) illustrate visually that our unified degradation form can predict degradation when training and serving in any precision. Plots (center, right) vary $P_\text{w}$ only, but fits in (left) include runs where $P_\text{a}, P_\text{kv}$ are also jointly varied.

> **Finding 3.** When $N, D, P$ are optimized jointly, compute-optimal pretraining precision is independent of compute. 16-bit has many unnecessary bits, and 4-bit requires increasing the model size disproportionately to maintain loss scaling. Our fits imply that 7-8 bits are compute-optimal. In contrast, when $N$ is fixed in advance, such as when training a model family on similar data, $P^*(C) \propto \log C$. This suggests that for models that will be significantly overtrained, higher precision during training may be compute-optimal.

## 5 A UNIFIED SCALING LAW FOR PRECISION

In this section, we combine the two scaling laws presented into a unified functional form that predicts both training/post-training effects, including interactions between the two. We now treat $\delta_\text{PTQ}$ as a function $\delta_\text{PTQ}(N, D, P_\text{train}, P_\text{post})$ rather than just $\delta_\text{PTQ}(N, D, P_\text{post})$ as we did earlier in Section 3. We find two competing effects at play when predicting $\delta_\text{PTQ}$, but **overall, models trained in lower precision are more robust to post-train quantization in the sense of incurring lower degradation.**

**Two competing effects at play during post-train quantization.** Intuitively, training any of $P_\text{w}, P_\text{a}, P_\text{kv}$ in low precision forces the model to learn weights that are robust to "quantization noise," so they degrade less under PTQ. However, the reduced $N \mapsto N_\text{eff}$ implies that models trained in low precision will degrade *more* because $\delta_\text{PTQ}$ increases with $N^{-\gamma_N}$ as we found in Section 3. We call this second effect the "overtraining" effect. In practice, the first "robustification" effect wins out, so that models trained in lower precision overall degrade *less* when post-train quantized. We confirm using $N_\text{eff}$ rather than $N$ to predict degradation given various training precisions leads to a substantially stronger fit in Figure 21(top left, top center), to verify the competing overtraining effect.

**Modifying $\delta_\text{PTQ}$ to account for training precision.** We assume training precision is strictly greater than inference precision, and define degradation as identically zero if they are equal. We begin by studying how degradation scales with just weight-precision during training, $P_\text{w}$.

Consider Figure 7(center). We fix $(N, D)$ and each cell of the heatmap represents the empirical degradation $\delta_\text{PTQ}(P_\text{w}, P_\text{post})$. We observe that degradation very quickly increases to its exponentially large value from Section 3 if there is any gap between training and inference-time precision. This

---

parameters in 16-bit and one with $2N_0$ parameters in 8-bit, the latter has higher $N_\text{eff}$ with our $\bar{\gamma}$, so will reach a lower pretraining loss on the same data with the same training/inference costs.

motivates modifying our initial functional form fitted in Section 3 to

$$\delta_{\text{PTQ}}(N, D, P_{\text{w}}, P_{\text{post}}) = C_T e^{-P_{\text{post}}/\gamma_{\text{post}}} \underbrace{\left(\frac{D^{\gamma_D}}{N_{\text{eff}}^{\gamma_N}}\right)}_{\text{Overtraining effect}} \underbrace{[1 - e^{-C_{\text{w}}(P_{\text{w}} - P_{\text{post}})}]}_{\text{Robustification effect}} \tag{8}$$

where $C_{\text{w}}$ is the only new fitted value. Then, we can extend this to include the precision effects of activations/attention in the natural way:

$$\delta_{\text{PTQ}}(N, D, P_{\text{w}}, P_{\text{a}}, P_{\text{kv}}, P_{\text{post}}) = C_T e^{-P_{\text{post}}/\gamma_{\text{post}}} \left(\frac{D^{\gamma_D}}{N_{\text{eff}}^{\gamma_N}}\right) \prod_{\text{x} \in \{\text{w,a,kv}\}} [1 - e^{-C_{\text{x}}(P_{\text{x}} - P_{\text{post}})}] \tag{9}$$

We measure the fit to the data of such a functional form in Figure 7, and find a strong fit with $R^2 = 0.90$ on over 1000 data points (each of 465 pretraining runs post-train quantized to multiple precisions).

**An interpretable, unified functional form.** Now we simplify and interpret the resulting functional form. Consider training with only weights in low precision and take $C_{\text{w}} = 1$ for illustrative purposes so we can simplify Equation 9. Denote $\sigma_{\text{tr}}^2 := e^{-P_{\text{w}}/\gamma_{\text{w}}}$ as "training noise" reflecting the decrease in effective parameter count due to training weights in lower precision. Then, Equation 9 simplifies to

$$\delta_{\text{PTQ}}(N, D, P_{\text{train}}, P_{\text{post}}) = C_T \underbrace{(\sigma_{\text{PTQ}}^2 - \sigma_{\text{tr}}^2)}_{\text{Robustification effect}} \cdot \underbrace{\left(\frac{D^{\gamma_D}}{N_{\text{eff}}^{\gamma_N}}\right)}_{\text{Overtraining effect}} \tag{10}$$

which we note is the intuitive modification one might make to the form of the initial post-training quantization degradation we fitted in Section 3, in Finding 3.1, with a small competing effects factor from $N_{\text{eff}}$ pushing in the opposite direction. *It cleanly reflects the intuition that models are robustified to PTQ noise to the extent they were trained with similar noise.*

> **Finding 4 (Unified Scaling Laws).** Modeling low-precision effects during pretraining as independent and multiplicative noise that accumulates, and including post-training quantization degradation, the predicted loss for a language model with $N$ parameters, trained on $D$ tokens, with training precision $P_{\text{w}}, P_{\text{a}}, P_{\text{kv}}$ to end-time weight-precision $P_{\text{post}}$, can be predicted as
>
> $$L(N, D, P_{\text{w}}, P_{\text{a}}, P_{\text{kv}}, P_{\text{post}}) = AN_{\text{eff}}^{-\alpha} + BD^{-\beta} + E + \delta_{\text{PTQ}} \tag{11}$$
>
> where $\delta_{\text{PTQ}}(N, D, P_{\text{w}}, P_{\text{a}}, P_{\text{kv}}, P_{\text{post}})$ is in general as in Equation 9 and $N_{\text{eff}}(N, P_{\text{w}}, P_{\text{a}}, P_{\text{kv}})$ as in Finding 4.2.

## 6 CONCLUSION AND LIMITATIONS

We find that the common inference-time technique of post-train quantization can incur large degradation at very high data budgets, demonstrating a striking example of how more pretraining compute does not always imply stronger models at inference-time. Seeking better data scaling, we study quantization-aware and low precision training. We find that parameters and bit precision are well modeled as interchangeably controlling an "effective parameter count" of the model allows us to predict finite-precision loss effects accurately during both training and inference.

There are limitations to our analysis. First, we use a fixed architecture throughout to examine the effects of precision, parameters, and tokens in a controlled manner. In contrast, low precision training often involves architectural tweaks (Ma et al., 2024; Zhu et al., 2024) that can close much of the gap from a vanilla full precision model. Second, while compute costs do scale linearly with precision, the gains from halving precision are usually less than 2x due to systems overhead. Third, we only consider loss scaling without downstream model evaluations. We emphasize that the trends we find aim to be suggestive rather than prescriptive, and hope future work can more comprehensively examine these effects at larger model scale. In all, we find that the effects of precision on loss are predictable and consistent, with important and surprising implications.

## 7 ETHICS STATEMENT

We study the efficient training of language models, and as such do not see any new ethical concerns arising as a result of our work.

## 8 ACKNOWLEDGEMENTS

Tanishq Kumar thanks Tim Dettmers, Chris De Sa, Neil Band and Luke Bailey for helpful comments and discussion, as well as Ludwig Schmidt for spotting an early typo. Blake Bordelon is supported by a Google PhD Fellowship. Cengiz Pehlevan is supported by NSF grant DMS-2134157, NSF CAREER Award IIS-2239780, and a Sloan Research Fellowship. This work has been made possible in part by a gift from the Chan Zuckerberg Initiative Foundation to establish the Kempner Institute for the Study of Natural and Artificial Intelligence. Aditi Raghunathan acknowledges support from AI2050 program by Schmidt Sciences (Grant G2264481), Google Research Scholar, Apple, NSF, Cisco.

We gratefully acknowledge the support of NIH under No. U54EB020405 (Mobilize), NSF under Nos. CCF2247015 (Hardware-Aware), CCF1763315 (Beyond Sparsity), CCF1563078 (Volume to Velocity), and 1937301 (RTML); US DEVCOM ARL under Nos. W911NF-23-2-0184 (Long-context) and W911NF-21-2-0251 (Interactive Human-AI Teaming); ONR under Nos. N000142312633 (Deep Signal Processing); Stanford HAI under No. 247183; NXP, Xilinx, LETI-CEA, Intel, IBM, Microsoft, NEC, Toshiba, TSMC, ARM, Hitachi, BASF, Accenture, Ericsson, Qualcomm, Analog Devices, Google Cloud, Salesforce, Total, the HAI-GCP Cloud Credits for Research program, the Stanford Data Science Initiative (SDSI). Benjamin F. Spector is supported by a Hertz Fellowship. The U.S. Government is authorized to reproduce and distribute reprints for Governmental purposes notwithstanding any copyright notation thereon. Any opinions, findings, and conclusions or recommendations expressed in this material are those of the authors and do not necessarily reflect the views, policies, or endorsements, either expressed or implied, of NIH, ONR, or the U.S. Government.

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

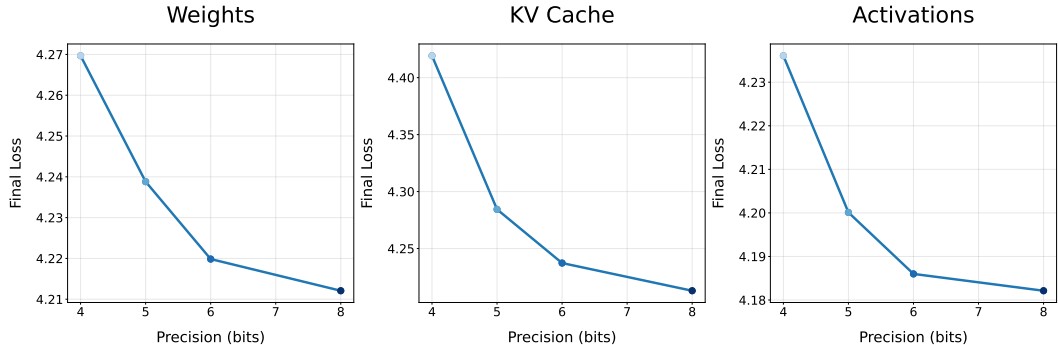

Figure 8: $L(P_\mathrm{w}), L(P_\mathrm{a}), L(P_\mathrm{kv})$ for ablated hyperparameters, $N = 30$M, $D = 1.5$B. We can see the trends persist, where the first few bits reduce final val loss significantly, with diminishing/saturating returns quickly setting in at higher precision. We do not fit constants on these ablated runs.

## A  HYPERPARAMETER DETAILS AND ABLATIONS

We launch over 20 runs for each $(N, D)$ combination to study scaling in precision, trained and validated on the common crawl split of the Dolma dataset (Soldaini et al., 2024). We use a standard causal Transformer++ implementation: SwiGLU activations (Shazeer, 2020), RoPE embeddings (Su et al., 2021), RMSLayerNorm, Adam $\beta$ values of $(0.9, 0.95)$. We adopt a cosine learning rate schedule with 10% warmup period and peak learning rate of 6e-4 for the smallest model and learning rates scaled with width and depth according to depth-$\mu$P for the larger models (Yang et al., 2022; Bordelon et al., 2023). We use a sequence length of 1024 and batch size of 256 throughout, with Adam $\epsilon$ 1e-15, following (Wortsman et al., 2023b). We use weight decay of $0.1$, as (Ahmadian et al., 2023) find some results in the quantization literature may be artifacts of insufficient weight decay. We follow (Ma et al., 2024) in including a LayerNorm before projections because they find it is important for low precision training to be stable. These are the hyperparameters and settings used for the main scaling law experiments.

To check robustness, we then ablate these hyperparameter choices, with results in Figure 8. In our ablation we use a sequence length of 512 with batch size 128, weight decay of 1e-3, Adam $\epsilon$ of 1e-10, a peak learning rate of 1e-4 and a warmup period of duration 3%. We train models with these alternative hyperparameters at various weight, activation, and KV cache precisions. We train and val on C4 (Raffel et al., 2020; Dodge et al., 2021) instead. Though these ablations are at rather smaller scale due to compute constraints, the loss curves follow the same trends – rapid decrease in final loss with an initial increase in precision from 4 bits, then diminishing returns as we approach higher precision – as in the main text, suggesting the trends are robust to hyperparameter choices.

## B  ADDITIONAL RELATED WORK

**Efficient training and inference**  Low precision has been key to improving the efficiency of training and using LLMs (Micikevicius et al., 2017; Shoeybi et al., 2019; Wortsman et al., 2023a; Zhu et al., 2023). Prior works generally study either precision during training (Courbariaux et al., 2014; Dettmers et al., 2024; 2021; Sun et al., 2020; Liu et al., 2023) or the effects of changing the precision after training (post-training quantization) (Frantar et al., 2022; Lin et al., 2024a; Dettmers et al., 2022; Xiao et al., 2023; Sheng et al., 2023; Dettmers et al., 2023). In this work we study both, the precision during training and after, and unify them from a scaling perspective. Other important works include recent popular work on quantization-aware-training (Ma et al., 2024) where weights are quantized to extreme precisions (ternary) on the forward pass during training. This work is consistent with ours in that they can quantize weights so aggressively because weights are less sensitive than activations or KV cache. Further, while we use a fixed architecture throughout to maintain a controlled comparison across precision, they use a nonstandard architecture, learning rate, and weight decay schedule specifically to make training with ternary weights stable.

**Large language models and scaling** By scaling up the transformer architecture (Vaswani et al., 2017) a variety of large language models have been proposed (Brown et al., 2020; Rae et al., 2021; Touvron et al., 2023a;b; Dubey et al., 2024; Le Scao et al., 2023; Muennighoff et al., 2022; 2024b; Groeneveld et al., 2024; Jiang et al., 2023; Zhang et al., 2022; Allal et al., 2023; Li et al., 2023; Lozhkov et al., 2024; Luukkonen et al., 2023; Bai et al., 2023; Chowdhery et al., 2023; Team et al., 2023; Üstün et al., 2024; Deitke et al., 2024). To improve our understanding of these models, various works have investigated their scaling properties (Ruan et al., 2024; Allen-Zhu & Li, 2024; Hägele et al., 2024). Many aspects are relevant to scaling including the architecture (Tay et al., 2022a; Krajewski et al., 2024; Tao et al., 2024; Clark et al., 2022; Tay et al., 2022b; Scao et al., 2022; Peng et al., 2024), the modalities considered (Aghajanyan et al., 2023; Alabdulmohsin et al., 2022; Cherti et al., 2023), the performance metrics (Wei et al., 2022; Srivastava et al., 2022; Isik et al., 2024), the data composition (Li et al., 2024; Liu et al., 2024; Albalak et al., 2024) and data repetitions (Muennighoff et al., 2024a). Our work analyzes one such aspect, which is key to better scaling: the numeric precision during and after training.

## C ALTERNATIVE FUNCTIONAL FORMS

There are several plausible functional forms to try a priori. The key junctions are whether a form is 1) additive or multiplicative and 2) interacts with parameters/data or is independent, 3) a power law or exponential. We try a variety of combinations of these three and find the formulation in the main text one of the best fits, notably with the fewest fitted parameters. We emphasize that several fitted forms are likely to be reasonable fits to the data, and an important desiderata for choosing a functional fit is interpretability. Several scaling law papers find multiple fits plausible in terms of predictive power (Muennighoff et al., 2024a; Kaplan et al., 2020), and ultimately make a decision based on interpretability.

We make these fit choices on sweeps of the form $L(N, D, P_W)$ and discuss alternatives to the decomposition/factorization to account for activations and KV cache in Appendix Section M, which assumes an effective parameter count formulation. In this section, a power law refers to a term of the form $C_w \cdot P^{-\alpha_w}$ where $C_w, \alpha_w$ are fitted. In general, we find modeling precision effects with power law fits on their own causes the fitted constants $A, B$ to blow up, whereas this does not happen with exponential fits, suggesting the power law does not change sharply enough to match the change in loss induced by precision. We note that while fitting parameters using a double notion of effective parameters *and* effective data leads to a slightly better fit, it requires more fitted parameters so we stick with the $N_{eff}$ formulation for simplicity and interpretability. When choosing between fits we validate on held-out data and the $R^2$ values below reflect the fit on the held out data. This is in contrast to our plots in the main text, where we have chosen a functional form and we fit and plot on the same data, as is standard in scaling laws (Muennighoff et al., 2024a).

| Functional Form | Val $R^2$ | Number of Fitted Parameters |
|---|---|---|
| $N_{eff}$ | 0.82 | 3 |
| Additive/independent power law | 0.71 | 2 |
| $D_{eff}$ | 0.74 | 3 |
| $N_{eff}$ and $D_{eff}$ (tied) | 0.79 | 3 |
| $N_{eff}$ and $D_{eff}$ (not tied) | 0.84 | 4 |
| Multiplicative power law, $N, P$ | 0.75 | 2 |

Table 1: Comparison of Functional Forms with $R^2$, and Number of Fitted Parameters

## D QUANTIZATION IMPLEMENTATION DETAILS AND TYPES

Two canonical types for neural network quantization are floating-point (FP) and integer (INT) quantization. Despite their differences in representation, we hypothesize the scaling behavior between floating-point and integer quantization can be described by similar functional forms, where 1(b) provides preliminary evidence for this.

## D.1 INTEGER QUANTIZATION AND IMPLEMENTATION DETAILS

In integer quantization, continuous values are mapped to discrete integer values. Typically, this is done by scaling the original values according to a fixed scale factor. Mathematically, for a real number $x$, the quantized integer value $x_{\text{int}}$ is computed as:

$$x_{\text{int}} = \left\lfloor \frac{x}{s} \right\rceil$$

where $s$ is the scaling factor, and $\lfloor \cdot \rceil$ denotes rounding to the nearest integer specified by the number of bits. The value can then be dequantized back to an approximate real value by multiplying by $s$:

$$x_{\text{dequant}} = s \cdot x_{\text{int}}$$

This process introduces quantization error, defined as the difference between the original value $x$ and the dequantized value $x_{\text{dequant}}$. The goal of quantization is to minimize this error while still reducing the precision. One can think of this as rounding to the nearest point on a uniform lattice. More complicated quantization schemes involve selecting the lattice points in a data or model-dependent manner. Integer quantization, as implemented, uses a fixed-point scaling based on the maximum absolute value of the tensor, and then scales the values within the range $[Q_n, Q_p]$, where $Q_n = -2^{(b-1)}$ and $Q_p = 2^{(b-1)} - 1$, with $b$ being the number of bits.

Integer quantization first rescales the inputs into the range specified by the number of bits by

$$s = \frac{Q_p}{\max(|x|)}$$

for tensor-based scaling, or

$$s = \frac{Q_p}{\max(|x|, \dim = k)}$$

for channel-based scaling. After scaling, the result is rounded to the nearest integer and then clamped to the range $[Q_n, Q_p]$. After matrix multiplication, the result is rescaled back into the original range. We quantize only the forward pass in this work, to ensure fair comparison between quantization-aware-training (weights only) and low-precision training (weights, activations, KV cache). This is because the backward pass is not usually quantized during quantization-aware-training (Ma et al., 2024), so comparing sensitivities of weights (forward only) to activations/KV cache (forward and backward) would not be a principled comparison. In production pretraining in low precision, the matrix multiplications on the backward pass are also quantized, leading to further compute savings. We leave a detailed analysis of how our observations change when accounting for the backward pass to future work. We use integer quantization throughout to fit our scaling laws for simplicity.

## D.2 FLOATING-POINT QUANTIZATION

Floating-point quantization is slightly more sophisticated, aiming to make a non-uniform lattice roughly matching the distribution of the weights, which are assumed to be Gaussian. A floating-point number is in general represented as:

$$x_{\text{fp}} = (-1)^s \cdot m \cdot 2^e$$

where $s$ is the sign bit, $m$ is the mantissa, and $e$ is the exponent. In floating-point quantization, both the mantissa and exponent are quantized to reduce the bit width. For exponent-mantissa allocations of bits and details of exponent bias, we follow the guidelines from (Micikevicius et al., 2022) and quantize weights per channel and activations per-tensor.

Making a full scaling law for floating-point quantization is more involved than our integer treatment, because the effects of scaling mantissa vs exponent bits are not the same. In contrast, in integer quantization, each additional bit simply causes us to round into a finer-grained lattice after rescaling, thereby reducing quantization error by a predictable amount. In floating-point quantization, altering the exponent affects the dynamic range, while altering the mantissa changes the precision within that range. This flexibility at once makes floating-point quantization more suitable for model training, but harder to analyze. We leave a commensurately detailed analysis of mantissa vs exponent – and more generally floating point – scaling to future work.

### D.3 HARDWARE DETAILS

Weight-only quantization can accelerate inference because software can be written to accommodate moving data between GPU parts (HBM-SRAM) in smaller units (types), so that a given bandwidth can move more data per second. This reduces memory (IO) bottlenecks that often dominate during inference, even with high-batch workloads. However, we emphasize that the type and therefore speed at which the GPU can do matrix multiplications in natively is determined by the hardware provider, so that even when $P_\text{w} = P_\text{a} = P_\text{qkv}$ (including queries), compute savings are only achieved when these correspond with both a bit-width and type that the GPU supports. We aim to study scaling in a fairly hardware-agnostic manner so that our work may be useful in the future, and make no claims about hardware details or optimality. We train all our models with fake (simulated) quantization on NVidia H100 GPUs to remain hardware agnostic, not taking advantage of any true low-precision computation. The only assumption is that when hardware does implement support for integer quantization, it is done in a way that involves some combination of rescaling and rounding, as is standard at the time of writing (Dettmers & Zettlemoyer, 2023; Dettmers et al., 2022; Wu et al., 2020; Jacob et al., 2018).

## E DERIVATIONS

### E.1 CRITICAL DATASET SIZE FOR PTQ

We seek a $D_\text{crit}$ that satisfies $\frac{\partial L(D_\text{crit})}{\partial D} = \frac{\partial \delta_\text{PTQ}(D_\text{crit})}{\partial D}$. Taking both derivatives for the functional forms presented in the main text and equating their opposing effects, we get the equation

$$BD_\text{crit}^{-\beta-1} = \gamma_D C_T N^{-\gamma_N} e^{-P_\text{post}/\gamma_\text{post}} D_\text{crit}^{\gamma_D-1} \tag{12}$$

which implies

$$D_\text{crit} = \left( \frac{\beta B N^{\gamma_N} e^{P_\text{post}/\gamma_\text{post}}}{\gamma_D C_T} \right)^{\frac{1}{\gamma_D+\beta}} \tag{13}$$

is the predicted point after which pretraining on more data can increase loss of a model that is post-train quantized. Note that this quantity explodes in $P$, so that a truly unreasonable amount of data is required for longer pretraining to be harmful at commonly used precisions (eg. 8-bit). However, we find that on overtrained models $D/N \gg 10^3$, these overtraining-degradation effects become nontrivial around 5-bits, and dominant below that.

### E.2 COMPUTE-OPTIMALITY CALCULATIONS

We set a constraint $C \propto NDP$ throughout. Working up to proportionality is essentially rescaling the compute constraint, so it doesn't affect the scaling trends we identify, which is our focus.

#### E.2.1 FIXED PRECISION COMPUTE OPTIMAL SCALING

Under fixed precision, the loss takes the form

$$L = u(P)AN^{-\alpha} + BD^{-\beta} \tag{14}$$

where $u(P) = [1 - e^{-P/\gamma}]^{-3\alpha}$ is a fixed constant. The compute optimal scaling when minimizing the loss over $N, D$ gives

$$L = u(P)AN^{-\alpha} + BC^{-\beta}N^\beta P^\beta \tag{15}$$

by replacing $D = \frac{C}{NP}$. Optimizing over $N$, we see that this is equivalent to the original chinchilla optimization problem but with $A \to Au(P)$ and $B \to BP^\beta$. Performing this optimization, we find

$$N^*(P,C) = \left( \frac{u(P)A\alpha}{BP^\beta \beta} \right)^{\frac{1}{\alpha+\beta}} C^{\frac{\beta}{\alpha+\beta}} \quad, \quad D^*(P,C) = \left( \frac{u(P)A\alpha}{BP^\beta \beta} \right)^{-\frac{1}{\alpha+\beta}} C^{\frac{\alpha}{\alpha+\beta}} \tag{16}$$

We can relate the above expressions to the original Chinchilla-optimal $N, D$ at full precision $N_{\text{Ch}}(C), D_{\text{Ch}}(C)$.

$$\frac{N^*(P,C)}{N_{\text{Ch}}(C)} \propto \left[1 - e^{-P/\bar{\gamma}}\right]^{-\frac{3\alpha}{\alpha+\beta}} P^{-\frac{\beta}{\alpha+\beta}} \text{ and } \frac{D^*(P,C)}{D_{\text{Ch}}(C)} \propto \left[1 - e^{-P/\bar{\gamma}}\right]^{\frac{3\alpha}{\alpha+\beta}} P^{\frac{\beta}{\alpha+\beta}} \tag{17}$$

### E.2.2 FIXED MODEL SIZE $N$

Now, we investigate the case where model size $N$ is fixed but precision and data are jointly optimized at fixed compute $C = NDP$. This optimization problem takes the form

$$L = u(P)AN^{-\alpha} + BD^{-\beta} \tag{18}$$

Under fixed compute, we have $D = \frac{C}{NP}$ so replacing the second term, we have

$$L = u(P)AN^{-\alpha} + BC^{-\beta}N^{\beta}P^{\beta} \tag{19}$$

where $N$ is a constant. We therefore have a single variable $P$ to minimize the above formula over

$$\frac{\partial L}{\partial P} = u'(P)AN^{-\alpha} + BC^{-\beta}N^{\beta} \beta P^{\beta-1} = 0 \tag{20}$$

First, we note that $u'(P)$ has the following form

$$u'(P) = -3\alpha[1 - e^{-P/\gamma}]^{-3\alpha-1} \times \frac{1}{\gamma}e^{-P/\gamma} = -\frac{3\alpha}{\gamma}e^{-P/\gamma} \times u(P)^{\frac{3\alpha+1}{3\alpha}} \tag{21}$$

We thus desire a solution to the implicit equation

$$\frac{3\alpha}{\gamma}e^{-P/\gamma} \times u(P)^{\frac{3\alpha+1}{3\alpha}} AN^{-\alpha} = BC^{-\beta}N^{\beta} \beta P^{\beta-1} \tag{22}$$

We now aim to find an approximate asymptotic relationship between $P$ and $C$ as $C \to \infty$. Taking a logarithm of both sides, we find (neglecting additive constants that are independent of $C, P$)

$$-(3\alpha + 1)\ln(1 - e^{-P/\gamma}) - \frac{1}{\gamma}P \approx -\beta\ln C \tag{23}$$

The correct dominant balance at large $C$ is to take $P^{\star} \sim \beta\gamma\ln C$, as can be verified numerically. With the constraint that $C = NPD$ we have that $D^{\star} \approx \frac{C}{N\beta\gamma\ln C}$.

### E.2.3 MINIMIZATION OVER $N$, $D$, $P$ WITH FIXED COMPUTE

Recall our three-way loss function is given as below. We separate $N_{\text{eff}}$ into terms involving $(N, P)$ explicitly here as it makes the math easier to follow.

$$L(N,D,P) = AN^{-\alpha}u(P) + BD^{-\beta} , \ u(P) = [1 - e^{-P/\gamma}]^{-3\alpha} \tag{24}$$

Under the constraint $C \propto NDP$, we can replace $D$ in terms of $C, N, P$ giving the loss expression

$$L = AN^{-\alpha}u(P) + BN^{\beta}P^{\beta}C^{-\beta} \tag{25}$$

$$\frac{\partial L}{\partial N} = -\alpha AN^{-\alpha-1}u(P) + \beta BN^{\beta-1}P^{\beta}C^{-\beta} = 0 \tag{26}$$

$$\frac{\partial L}{\partial P} = -3\alpha/\gamma AN^{-\alpha}u(P)^{\frac{3\alpha+1}{3\alpha}}e^{-P/\gamma} + \beta BN^{\beta}P^{\beta-1}C^{-\beta} = 0 \tag{27}$$

Multiplying the first equation by $N$ and dividing the second equation by it reveals that the optimal $P$ satisfies a compute-independent implicit equation

$$\frac{3}{\bar{\gamma}} u(P)^{\frac{1}{3\alpha}}e^{-P/\bar{\gamma}} = P^{-1}u(P) \tag{28}$$

This exercise reveals that the compute optimal strategy when allowed to jointly optimize $N, D, P$ is to choose a fixed precision that satisfies the above equation and then to scale up $N, D$ with the prescription in Appendix I.1.1.

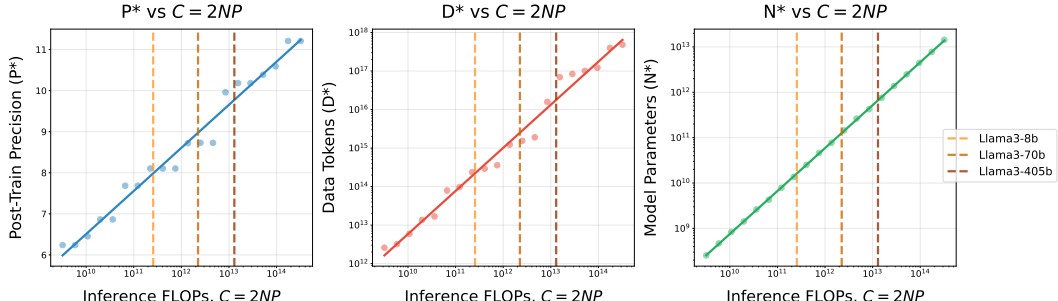

Figure 9: Numerically minimizing a model of inference-time costs with respect to $N, D, P$ after accounting for post-train-quantization degradation and its relation to overtraining.

### E.3 INFERENCE-TIME COST MODEL

For many, inference is the primary cost of training and serving models. Here, we present a preliminary analysis of an inference-time cost model. The key tension is that inference cost scales as $NP$, so that inference costs at a fixed pretraining loss can be reduce by either reducing model size (and overtraining more) or quantizing post-training

We will assume here that $P = P_{\text{post}}$ refers to the precision weights will be quantized to. In practice, inference costs may depend on the precision of the KV cache and activations to some extent as well, but we assume this for tractability of the following mathematical model, and to get a sense of how overtraining and post-train quantization concerns play out at inference-time. We can phrase this minimization problem in the following way.

$$\min_{N,D,P} L(N, D, P) = A N^{-\alpha} + B D^{-\beta} + C_T \frac{D^{\gamma_D}}{N^{\gamma_N}} e^{-P/\gamma} \text{ subject to } C = NP \qquad (29)$$

The system of first-order conditions that results from this constrained optimization problem is not in general tractable analytically, so we solve the above constrained optimization problem for $P^*(C), N^*(C), D^*(C)$ numerically via a simple grid search. We find that $N^*, D^*$ grow as a power law in $C$ while $P^* \propto \log C$. The clumping in points is an artifact of the numerics of the grid search; the fitted lines represent the loglinear (left) and loglog (middle, right) trends overall.

It might be surprising that $D^*$ is not taken to infinity since it does not appear in the cost function. The reason for this is because if it was, post-train degradation (the third term) would become large. It might also be surprising that $D^*$ changes with compute at all. The reason for this is because, once again, of the third term: as we allow more inference-time compute we use more $N$, and at a larger $N$ we can now tolerate a larger data budget for a given post-train quantization degradation, so being compute-optimal means taking advantage of this and training that larger parameter count on more data.

The intuition for why $P^* \sim \log C$ might be as follows. Consider a situation in which $P^*$ is independent of compute: the third term will come to be a bottleneck in loss as compute gets larger because $N, D$ are both being scaled as power laws in compute, and eventually the effect of $e^{-P/\gamma}$ will become non-negligible in comparison to the first two terms in the loss function. To continue decreasing loss at this point, we must make this term smaller at a rate commensurate with the other terms, which go as a power law in compute. Since precision is inside the exponential, this can be done by taking $P \sim \log C$. An important thing to note is that since we are ignoring pretraining costs here, the absolute values of predicted $D^*$ are much larger than would be realistically possible in any reasonably training regime, where pretraining costs do matter, even if less than inference costs. But the empirical trends in $N^*, P^*$ showcase how overtraining with post-train quantization in mind can outperform vanilla overtraining without accounting for its effects on post-train quantization.

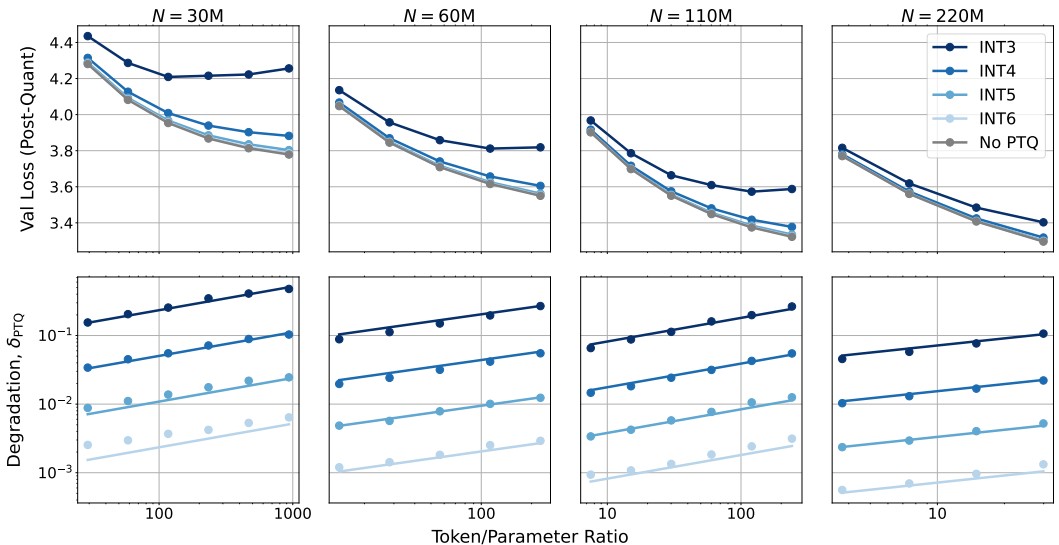

Figure 10: Replicating Section 3 results with AWQ.

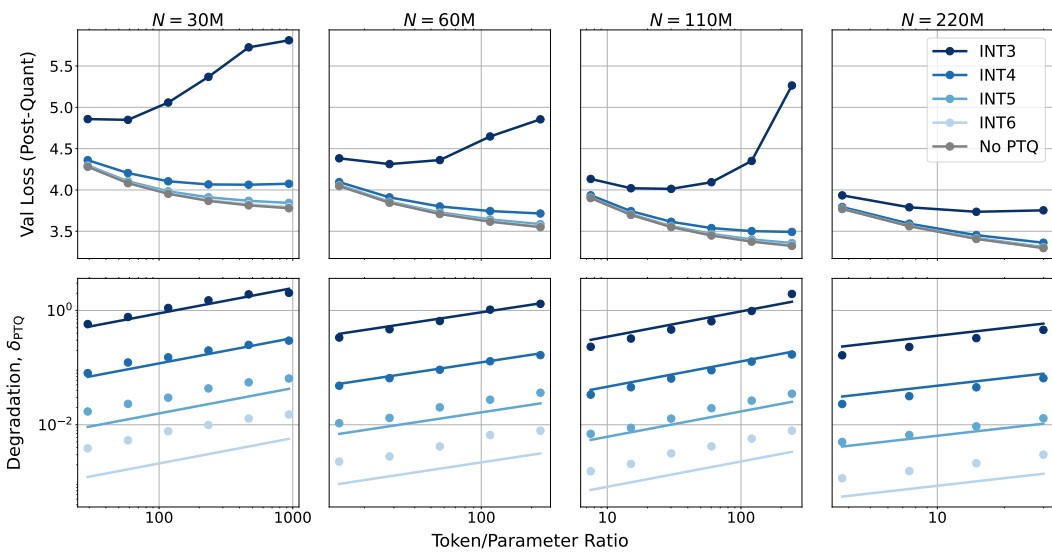

Figure 11: Replicating Section 3 results with RTN.

## F  REPLICATING PTQ SCALING WITH OTHER QUANTIZATION METHODS

Here we replicate the finding that post-train degradation due to post-train quantization increases with token/parameter ratio as $D^{\gamma_D}/N^{\gamma_N}$. We fit the same functional form as in the main text, but get slightly different values of fitted constants, as expect. We replicate on AWQ (Lin et al., 2023) and round-to-nearest quantization. The former is a modern and sophisticated technique, and the latter a simple and naïve approach to quantization. The fact they, as well as GPTQ in the main text, share the same failure modes suggests that poor post-training quantization data scaling should be the default expectation for any newly proposed PTQ technique.

## G  PTQ: LEARNING RATE SCHEDULE ABLATION

Here, we ablate our learning rate and schedule to use warmup with linear decay, as opposed to a cosine schedule, to check it is not an artifact of our choice of learning rate schedule. We do so

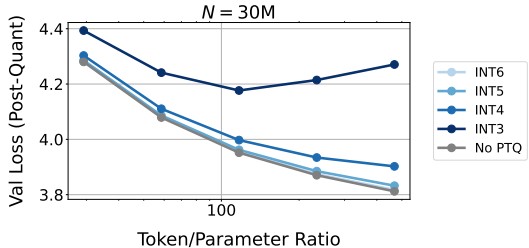

Figure 12: Linear LR Schedule Ablation

on our 30M model due to compute constraints, finding the degradation with token/parameter ratio persists, as expected.

## H    WHY DO LANGUAGE MODELS GET MORE SENSITIVE WITH OVERTRAINING?

This section is speculative.

**Sharpness.** A canonical line of work in optimization demonstrates that model sharpness increases during learning until it hovers at a maximal value (the "edge of stability") (Cohen et al., 2021; Gilmer et al., 2021), so that movement along the top Hessian eigenvector degrades loss by more throughout training. Though sharpness is formally a worst-case sensitivity, we conjecture similar results hold for average case, such as loss degradation induced by isotropic noise. It may be possible that sharpness during language model pretraining does not reach its maximal value for a long time, which is why sensitivity to noise monotonically seems to increase as $D/N \to \infty$ on realistic data budgets. Closely related is the largest eigenvalue of the neural tangent kernel (NTK) which captures the magnitude of the variance of the predictor under parameter noise. This quantity is known to empirically increase during training in a variety of settings, and is closely related to generalization guarantees (Nguyen et al., 2021; Atanasov et al., 2022).

**Hierarchical learning strategies become more sensitive throughout training.** Our expectation that overtrained language models may degrade more when quantized at inference-time is motivated in part by the following results. The hierarchical nature of learning is by now well understood in some toy settings: in (Abbe et al., 2021), it is shown that "staircase" polynomials of increasing degree are learned faster than high-degree monomials since neural networks combine existing features to learn new ones. In (Abbe et al., 2022) this result was strengthened to show that such hierarchical structure is both necessary and sufficient to learn sparse functions with SGD in two layer neural networks. In this setting, damage to features encoding lower-order polynomials affects all higher-order ones, so that such networks are increasingly sensitive to fixed feature noise throughout learning. Another result of a similar flavor is that of (Barak et al., 2022), who explicitly require high-precision gradients for sparse parity to be learned, since sparse parity is learned by the amplification of a small initial signal. If language models learn hierarchically, it is possible that the features that are learned late into overtraining as $D/N \to \infty$ are reliant on base features, so that noise harms the base features and therefore significantly damages higher-order features.

## I    GRANULARITY ABLATIONS

Here, we ablate our choice of quantization granularity (per-tensor vs per-channel) compared to the main text, where we do weights per-channel and activations per-tensor. Per-tensor quantization involves keeping one scalar to rescale all values in a tensor into the quantization codebook range, and per-channel means keeping a scalar per channel dimension; therefore, the latter is strictly more expressive and thus incurs lower quantization loss, than the former, at the cost of slightly more memory usage. Here, we ask: is the increased sensitivity of activations a result of them being inherently more sensitive, or due to the per-tensor design choice.

## Varying Quantization Granularity

Figure 13: Quantization granularity ablation: all combination of (training weight precision, training activation precision) × (per-tensor, per-channel). Dashed lines are per-channel and solid are per-tensor.

These results show that activations are generally more sensitive than weights, since their loss penalty at lower precision goes up faster even when granularity is kept fixed across the two. In fact, quantizing activations per-channel is almost as hard as quantizing weights per-tensor. This is consistent with a broad line of work in quantization finding that activations comprise the central difficulty in quantization (Dettmers & Zettlemoyer, 2023; Ma et al., 2024).

## J  MAIN FIGURE DETAILS

The model on the left is $N = 30$M parameters, chosen because we could train it to the highest token/parameter ratio given our compute budget. On the right we train a suite of models with $NP$ kept constants on 16B tokens (so that $C = \frac{6}{16}NDP$ is matched throughout under our cost model). We plot val loss on Dolma, as throughout the main text, and use floating-point (rather than integer) to make the pretraining claims as realistic as possible.

## K  NUMERICAL FITS

Following (Muennighoff et al., 2024a), we tie $\alpha = \beta$ so they do not become very different, though this is not required. Distinct $\alpha, \beta$ only add expressivity to the model and we have verified the plots look similar without tying. We also only use the full scaling law when specified in the text, since the law is developed piecewise through the text. For instance, Figures 3 and 4 solely fit Chinchilla with a substitution $N \mapsto N_{\text{eff}}(P_{\text{w}})$ because at that point $P_{\text{a}}, P_{\text{kv}}$ have not been introduced. Figures 5, 6, and 7 use our full scaling law, for instance to make predictions. We emphasize our numerical constants are unlikely to be useful because as (Hoffmann et al., 2022; Sardana & Frankle, 2023) show, fitted constant depend heavily on the architecture and dataset used, which differs from setup to setup. Rather, the trends we identify are the key findings. With that said, our fitted constants are as follows.

Note that we include biases in our exponent fits, for instance when modelling $N_{\text{eff}}$ as a saturating exponential, we find that the different parts of a model cause numerical instability at different values of low precisions, so even if they are the same functional form, they may be translated (left/right shifted versions) of eah other. For instance a fit of the form $e^{x/\gamma_x}$ in the main text is really computed with offset $e^{x/\gamma_x + n}$, but including biases everywhere clutters notation and obscures mathematical insight.

| Constant | Value |
|---|---|
| $A$ | 4.299e3 |
| $\alpha$ | 0.4965 |
| $B$ | 1.806e4 |
| $E$ | 2.7648 |
| $\gamma_{\mathrm{w}}$ | 2.6745 |
| $n_{\mathrm{w}}$ | 0.3037 |
| $\gamma_{\mathrm{i}}$ | 2.2102 |
| $n_{\mathrm{i}}$ | 1.4072 |
| $\gamma_{\mathrm{kv}}$ | 0.9578 |
| $n_{\mathrm{kv}}$ | 2.4185 |
| $C_T$ | 0.0598 |
| $\gamma_D$ | 0.5068 |
| $\gamma_N$ | 0.3439 |
| $\gamma$ | 0.5907 |
| $b$ | 1.1277 |

Table 2: Fitted constants and their values

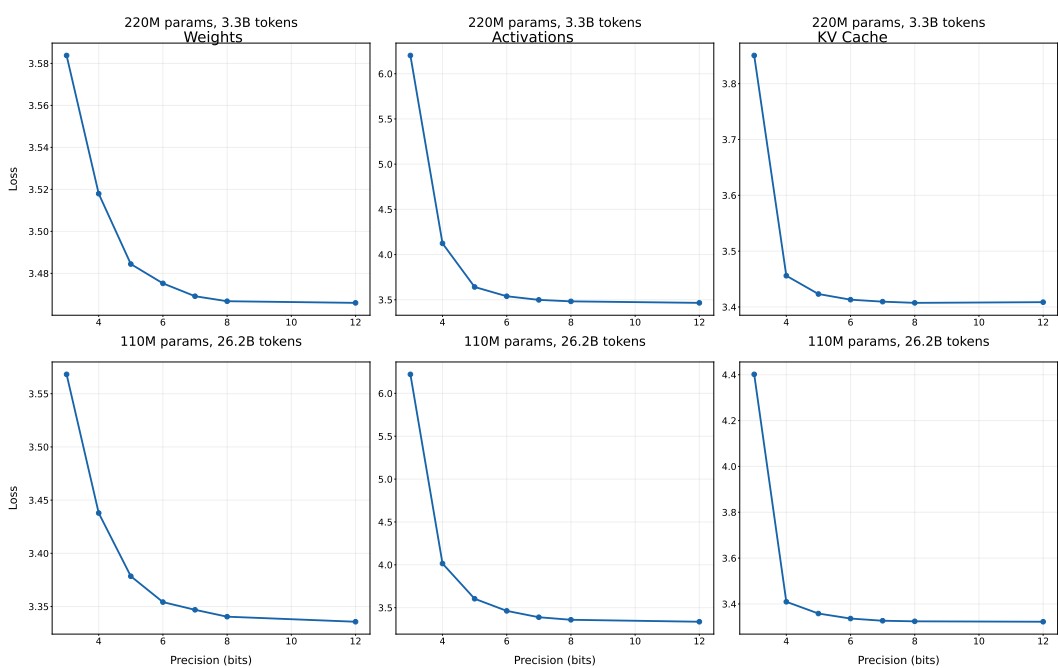

Figure 14: Sweeping $L(P)$ for the three model parts at various $N, D$.

## L  ARE WEIGHTS, ACTIVATIONS, AND KV CACHE EQUALLY SENSITIVE?

We find that training runs with $P_a \leq 3$ or $P_{\mathrm{kv}} \leq 3$ are not numerically stable, and often diverge, while $P_{\mathrm{w}} = 3$ is still well behaved. In particular, we find activations are more sensitive, though this could be because we quantize activations per-tensor and weights-per channel, rather than activations being inherently more sensitive. Consequently, we do not fit or validate on runs with activations or attention bits equal to 3. We leave a more detailed analysis of fine-grained sensitivity across layers and types of parameters to future work. The Figure below illustrates the empirical sensitivity by plotting $L(P)$ for the three quantities for various runs $(N, D)$.

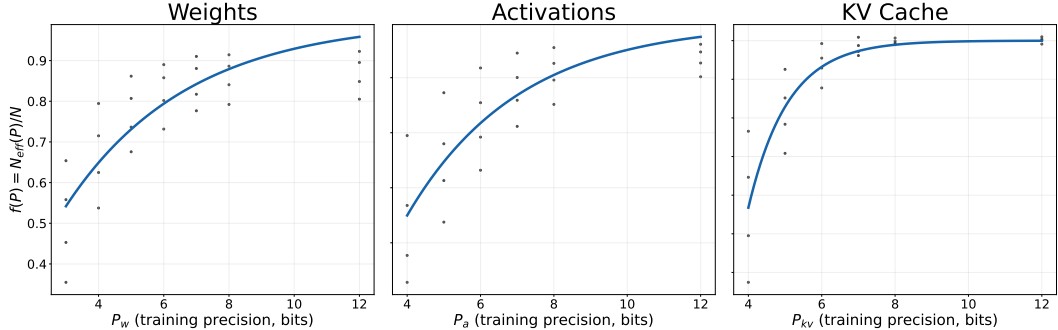

Figure 15: Plotting what $N_{\text{eff}}$ looks like empirically. Each black point is a pretraining run, mathematical details of what is plotted here in Appendix E. Blue lines are parametric fits of a saturating exponential.

## M EMPIRICAL $N_{\text{EFF}}$

Consider a model trained with some arbitrary $(N, D, P_{\text{w}})$. Assuming a Chinchilla function form with $N \mapsto N_{\text{eff}}(P_{\text{w}})$, we can write the difference between its loss and the loss of a full precision model as

$$L(N, D, P_{\text{w}}) - L(N, D, \infty) = A[N_{\text{eff}}^{-\alpha} - N^{-\alpha}]$$

as the terms involving $B, D, E$ cancel. Note that $N_{\text{eff}}(P_{\text{w}} = \infty) = N$ by construction. In practice, we use a BF16 model as the "infinite-precision" model, finding no real difference if we use an FP32 model or even a functional fit estimating $P_{\text{w}} \to \infty$ based on our integer quantization loss results. Our goal is to plot what $f(P)$ looks like where $N_{\text{eff}} = N \cdot f(P)$. Therefore, we can rearrange the above equation as follows

$$f(P) := \frac{N_{\text{eff}}}{N} = \frac{1}{N} \left[ \frac{L(N, D, P_{\text{w}}) - L(N, D, P_{\text{w}} = \infty)}{A} + N^{-\alpha} \right]^{-1/\alpha} \tag{30}$$

Then plotting this quantity using our fitted numerical values (See Appendix K) gives us the empirical tradeoff between precision and parameters. We can see that the tradeoff is quickly saturating in $P$ to a value near 1. While the functional form is the same for the three model parts, the fitted constants are different. For instance, runs with $P_{\text{a}} \leq 3$ or $P_{\text{kv}} \leq 3$ often diverged, and this was not the case with weight precision. Further, we can see that the KV cache is not sensitive to quantization at higher bit value, but very quickly becomes sensitive around 4-5 bit precision.

Then as far as the joint functional form for $N_{\text{eff}}(P_{\text{w}}, P_{\text{a}}, P_{\text{kv}})$ is concerned, we acknowledge that alternative factorizations that do not decompose the model into weights, activations, and KV cache, may have an equally good fit. For instance, decomposing the weights term into a product of layer-wise effects has a reasonable fit though introduces more parameters, and a more coarse-grained version may not decompose the model into parts at all, but only consider tied precisions. We choose this factorized form because QAT considers weights only, and activations and attentions are the two other things that must then be kept in low precision to see compute gains. Since practitioners often care about KV cache on its own, we chose to decompose "activations and attention" as "activations and KV cache." We emphasize that our main point is not that *this* factorization is objectively correct, but in observing that such a *factorization* that assumes approximate independence is *possible in the first place*.

## N FLOATING-POINT EXPERIMENTS

The key difference between floating point and integer type is that the former allocates some bits to the *exponent* representation and some to the *mantissa*, and these bits play different roles, unlike in integer type where every bit plays the same role in making the quantization lattice uniformly more fine-grained. We hypothesize that if exponent and mantissa bits are scaled jointly (ie. increase

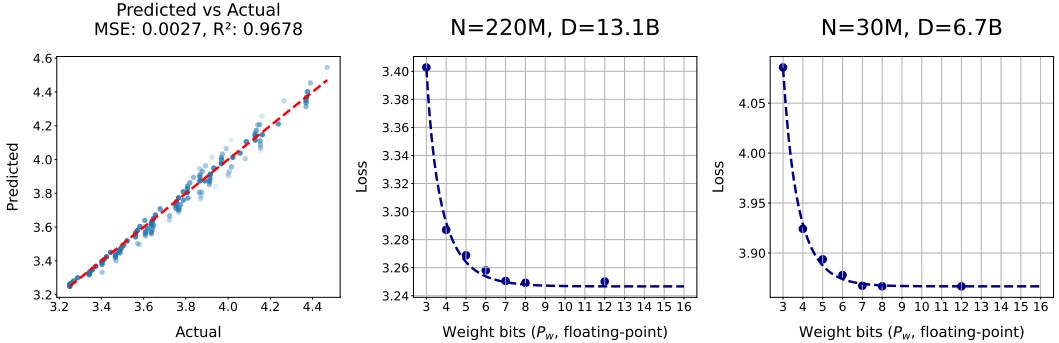

Figure 16: Fitting an effective parameter form to floating-point precision for weight training. (Left) involves checking quality of fit on 140 training runs in floating point precision for weights during training.

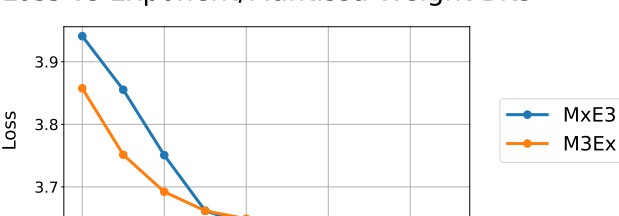

Figure 17: Exponent-mantissa bit allocation sweep. We can see the two types of bits have different scaling behavior, but both fit the saturating form where the first few bits reduce loss a lot, with diminishing returns after that.

together as total bit count does), the overall trend will still be predictable with a functional form like ours. To test this, we fit a parametric form like Equation 3 with the constants $A, B, E, \alpha = \beta$ listed in the table. The overall fit results in values of $\gamma_w = 2.8111$ and an exponent bias of $b = 0.1240$, showing the functional form is still a good fit to the data, even for floating point, under reasonably standard bit allocation schemes between mantissa and exponent. On the middle and right, we fit the same parametric form for particular values of $(N, D)$ and visualize the quality of the resulting predictions.

We use bit allocations of E2M0, E3M0, E4M1, E3M2, E4M2, E5M2, and E5M6 for 3, 4, 5, 6, 7, 8, 12 bits, respectively, with one sign bit throughout. Since exponent and mantissa bits play in general different roles (ie. the effect of a bit on loss and dynamics depends a lot on whether it comes from the mantissa or exponent in floating point), we expect our functional form does well here because mantissa and exponent allocations both increase jointly as precision rises, so overall the trends are predictable in a similar way. We check directly the role of the two by sweeping ExM3 and E3Mx directly, confirming this intuition. This suggests one route for making fine-grained fits for general arbitrary ExMy combinations is to decompose the effects of mantissa and weights, for instance a form like $N_{\text{eff}}(P_{\text{w, m}}, P_{\text{w, e}}, N)$. Since this is not needed for standard bit allocation choices as we can see in Figure 16, we do not delve into this complexity.

## O  ADDITIONAL PLOTS

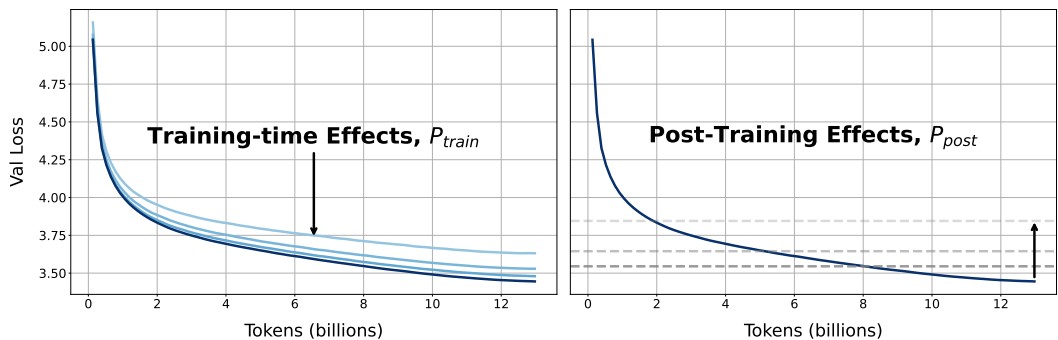

Figure 18: Illustration of what finite-precision effects during training and inference look like on learning curves.

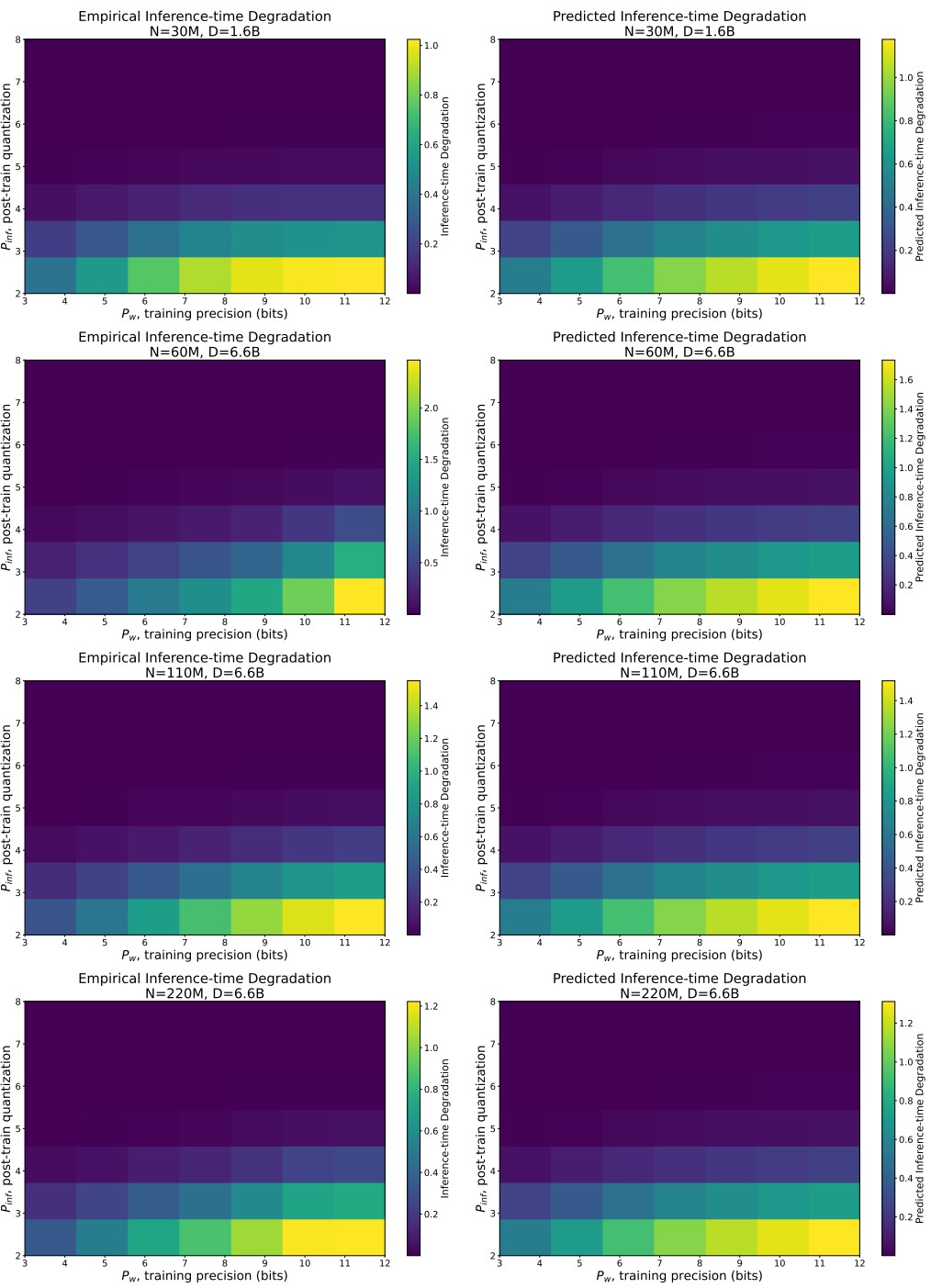

Figure 19: Predicted vs actual $\delta_{\text{PTQ}}$ for several $N, D$.

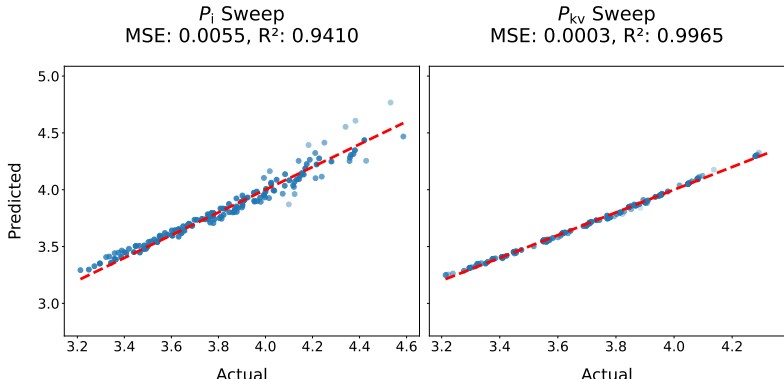

Figure 20: Marginal sweeps for precision of activations and KV cache, along with predictions from an $N_{\text{eff}}$ functional form analogous to Equation 3 fitted from scratch.

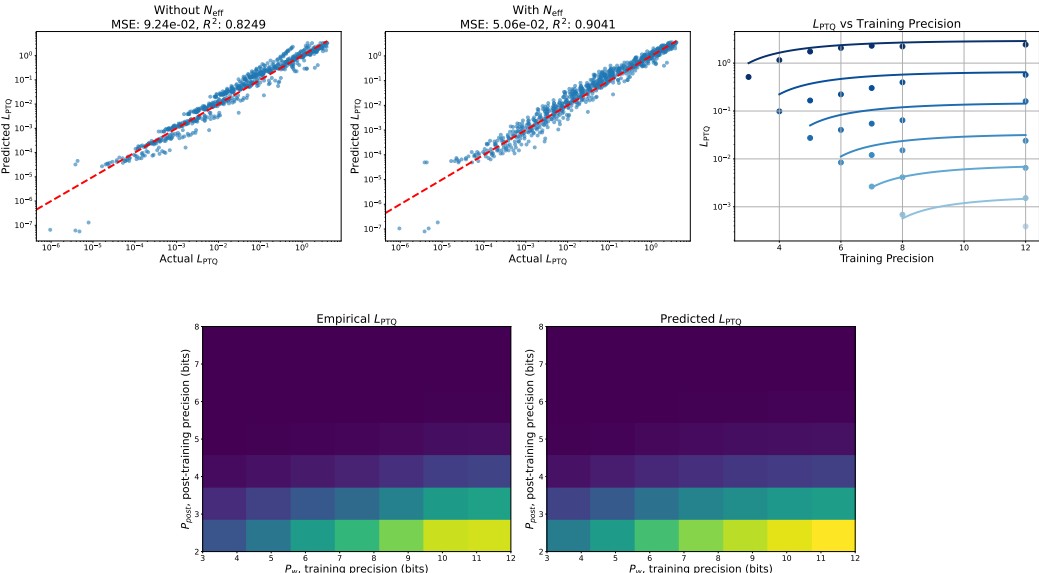

Figure 21: Combined plots for predicting degradation. (a) and (b) illustrate different fitting approaches to model degradation, demonstrating a stronger fit when $N \mapsto N_{\text{eff}}$ is used. (c), (d) (e) illustrate our unified degradation form can predict degradation when training and serving in any precision. Plots (c-e) made for varied $P_{\text{w}}$, but fits in (a) and (b) include runs where $P_{\text{a}}, P_{\text{kv}}$ are also jointly varied.

