# OpenReview forum: "Scaling Laws for Precision"
_ICLR.cc/2025/Conference — ICLR 2025 Oral_

### Official Review · Reviewer_heoa · 2024-10-31

**Soundness:** 4
**Presentation:** 4
**Contribution:** 3
**Rating:** 8
**Confidence:** 4

**Summary:**

This paper propose the scaling laws for precision through replacing the N in the original Chinchilla with the effective parameter count $N_{eff}$ and adding the post-training effects.

**Strengths:**

1.	The proposed scaling law unify the post train quantization and quantized training into a single functional form.
2.	The finding in the section 4.3 is inspired and the conclusions are consistent with usual experience and give a theoretical explanation.
3.	The experiment is adequate and reasonable and the paper is well written.

**Weaknesses:**

1.	The paper use the $N(1-e^{P_{w}/\gamma_{w}})$ to fit the left in the figure 3. But I think the power law is the most commonly used in all kinds of scaling law form. I suggest the author could compare the exponential with power law like $N(1- A*P_{w}^{\alpha})$.

**Questions:**

As shown in above.

---

> ### Author Response · Authors · 2024-11-20
> **Official Response by Authors**
>
> Thank you for the review! You’re right that power laws are common in scaling laws. For this reason, we compare directly to the power law functional form you point out in Appendix C, finding it has a weaker fit to the data than the exponential form we use in the main text. Note that an exponential model might also be reasonable because of the information-theoretic observation that $k$ bits can encode $2^k$ states, so that one may hypothesize that the first few bits contribute a large amount to effective parameter count, with quickly diminishing returns after that. But your point that this is an important baseline comparison is a good one.
>
> We’ve also made a number of significant improvements to the manuscript and uploaded a new version with new Appendix Sections in blue:
> - We’ve **added preliminary experiments studying quantized training with FP instead of INT, involving over 100 new pretraining runs.** Going beyond our functional form, we examine how loss scales with FP bits under non-standard mantissa vs exponent bit allocations. See Appendix N.
> - We’ve **added a new inference cost model studying the optimal choice of overtraining vs post-train quantization for workloads where inference is the dominant cost.** This complements our training cost analysis to now provide insights for practitioners who care mostly about inference. It can be found in Appendix E.3.
> - We’ve **added experiments in Appendix I testing the effect of per-tensor vs per-channel quantization** on scaling behavior with weight vs activation precision, finding that while granularity can explain part of the difference between weights and activations, activations are indeed inherently more sensitive to quantization at the same quantization granularity.
> - We’ve added minor improvements throughout the manuscript: reformatting some plots, changing our validation shard, rewriting some paragraphs for clarity, and including more implementation details.
> - We’ve **added ablations for post-train-quantization**: we reproduce our degradation results using AWQ, a modern post-train quantization method, as well as when we train with a linear learning rate schedule instead of cosine. The behavior persists under both ablations. See new Appendix F and G.
>
> We think these additions improve the manuscript substantially. If there’s any experiments that would further improve your assessment of our work, feel free to let us know. Many thanks for the positive review.

---

### Official Review · Reviewer_TyGC · 2024-11-01

**Soundness:** 3
**Presentation:** 3
**Contribution:** 3
**Rating:** 8
**Confidence:** 3

**Summary:**

This paper explores how precision -- specifically, low-precision training and inference -- affects the performance and compute cost of large language models. The authors propose new "precision-aware" scaling laws to predict the degradation in model performance when trained or quantized at different precision levels. Their work is motivated by the increasing trend toward low-precision training, driven by the need to reduce computational costs while maintaining model quality. While previous research has focused on scaling laws that balance model size and dataset size (for example Hoffmann et al. Chinchilla scaling laws), these do not account for the role of precision. The authors argue that precision is a critical factor that influences both compute efficiency and model performance, especially as hardware evolves to support lower precisions. They aim to fill this gap by developing scaling laws that incorporate precision as a third variable alongside model size and dataset size.

**Strengths:**

- The paper introduces a new dimension to the well-established scaling laws by incorporating precision as a critical factor. This is an important contribution because most prior work focused on model size and dataset size without considering precision, which is increasingly relevant due to hardware advancements supporting lower-precision computations. By doing so, the authors offer a more comprehensive framework for understanding and optimizing model performance under different training and inference conditions.

- The authors fit on over 465 pretraining runs across different precisions (3-bit to 16-bit) and sizes (up to 1.7 billion parameters), providing a robust dataset to validate their proposed scaling laws. The empirical results are consistent with the theoretical predictions, achieving high R^2 values (e.g., R^2 = 0.97 for post-training quantization degradation).

- The paper offers actionable insights into how low-precision training can be compute-optimal, particularly in scenarios where hardware constraints or cost considerations are paramount. For example, it shows that training larger models at lower precision can sometimes be more efficient than using higher precision, which is a valuable insight for practitioners looking to optimize both performance and computational costs.

**Weaknesses:**

- While the paper focuses extensively on integer-type precisions (e.g., 3-bit, 8-bit), it does not explore floating-point types like FP8 or BF16 in as much depth. Given that floating-point formats are widely used in modern hardwares, this omission limits the generalizability of the findings to real-world applications where floating-point precision is common. This could limit the applicability of the scaling laws in environments where floating-point precision dominates, potentially requiring further research to adapt these findings.

- The experiments are conducted on specific hardware setups that support low-precision computations, such as GPUs optimized for integer-type operations. The fitted constants and trends may not generalize well across different hardware architectures or future technologies that handle precision differently. This may reduce the long-term relevance of the paper’s findings as hardware evolves.

- Maybe I'm missing this, but the paper suggests that compute-optimal precision is around 8 bits but does not deeply explore scenarios where precision drops below 4 bits (e.g., binary or ternary quantization). Given that future hardware may support even lower precisions, this limits the scope of the findings.

- While pretraining cost optimization is thoroughly explored, inference-time costs -- especially in real-time or latency-sensitive applications -- are not given as much attention. In many practical deployments, inference-time efficiency is more critical than pretraining cost savings. This imbalance might limit the practical applicability of some of the findings in scenarios where inference-time efficiency is more important than pretraining considerations.

**Questions:**

I already specified some of them above, but the questions are particularly as in the following:

- While the paper primarily focuses on integer-type quantization, you mention that floating-point quantization  is commonly used in practice, especially in pretraining. Can you elaborate on how your scaling laws might differ when applied to floating-point quantization?

- You mention in the paper that activations and KV cache are more sensitive to low precision than weights, particularly when precision drops below 4 bits. Could you provide more detailed insights into why activations and KV cache are more sensitive? Is this primarily due to the per-tensor vs per-channel quantization method, or are there other factors at play?

- Your experiments are conducted using specific hardware such as Nvidia H100 GPUs. How do you expect the scaling laws to generalize across different hardware architectures, especially those that may handle precision differently, for example future GPUs with native support for FP4 or binary/ternary quantization?

- Given that your largest model size is 1.7B parameters, do you anticipate any limitations or deviations from your scaling laws when applied to much larger models with hundreds of billions or trillions of parameters?

---

> ### Author Response · Authors · 2024-11-20
> **Official Response by Authors, Part 1**
>
> Thank you for the detailed feedback and close reading! We’ve added a lot more experiments and ablations to the manuscript directly in response to your excellent questions, which we believe strengthen the manuscript substantially. You can find them in new Appendix sections, which are written in blue in the updated manuscript, and we point to them in our response.
>
> - **Floating-point precision.**
>
> **We launched and analyzed over 100 new pretraining runs specifically to address this question (new Appendix N)**, in which we vary weight training precision now in floating point instead of integer type. In short, we find that our existing functional form works well, as evidenced by having a high validation $R^2$ value (predictive power). One important conceptual difference between floating point and integer type is that FP bits are subdivided into exponent and mantissa bits, so these new pretraining runs assume reasonable and standard bit allocations between exponent vs mantissa as we scale precision. Making similar predictions for nonstandard allocation choices may require adapting our functional form by subdividing precision terms in our scaling law into exponent and mantissa. As a first step towards this, we sweep exponent and mantissa bits on their own and show they do not have the same scaling, since they play different roles in the floating-point representation scheme. But this is a technicality, with any standard bit allocation scheme we find our functional form works out-of-the-box for the preliminary floating-point sweeps we have added, suggesting that our functional forms or variants of them may work across types.
>
>
> - **Extremely low (binary/ternary) precision.**
>
> Two things to say here.
>
> - Recent popular work on binary/ternary quantizes only weights. For instance, [Ma et al, 2024] leave activations in 8-bit. This is consistent with our findings that activations/KV cache are more sensitive than weights at extreme precision. The difficulty with extreme quantization lies in the activations/KV cache rather than the weights, as predicted by our Fig3 (left).
> - Even just training weights in binary/ternary requires many changes to the standard architecture/training recipe. Training a vanilla (causal Transformer++) language model in the manner we do with binary/ternary weights, for instance, is unstable: the loss will diverge. For this reason, [Ma et al, 2024] make substantial modifications to their training setup, for instance using a custom learning rate and weight decay schedule. For us, changing architecture or training recipe across precisions would make it an unfair comparison across precisions since these changes made for stable low-precision training often do not work well compared to a standard training recipe when things are at high precision.
>
> So the short answer to your question is that there is no obvious principled way to compare to such extreme precision weights in the setup we’re interested in studying since it requires nonstandard architectural/training changes that would not be used for usual training at non-extreme precisions. We’ve **added discussion on this point in Appendix B,** since readers may also have this question.
>
> - **Weight vs activation sensitivity – per-tensor vs per-channel.**
>
> We’ve **added experiments in Appendix I to test this:** we sweep weight precisions and activation precisions during training using both in per-tensor and per-channel at a fixed model size and data budget. We see that the degradation when going from per-channel to per-tensor is larger for activations than weights, and that activations per-channel are comparable to weights per tensor. This suggests that the per-tensor vs per-channel implementation detail accounts for some, but not all, of activations’ increased sensitivity in our main text as compared to weights. This is consistent with much existing work in quantization that finds activations to be harder to quantize than weights themselves.
>
> - **Inference-time cost analysis.**
>
> **This is a great idea. We’ve added an inference-time cost analysis in Appendix E.3.** The key tradeoff is that to get cheap inference at a fixed desired loss, you can 1) overtrain small models, 2) train a larger model and post-train quantize, 3) do some combination. Which is optimal if a practitioner's main concern is inference costs? We set up a cost model and solve it numerically, finding that in short, models that are extremely overtrained should be made slightly larger than quantized more aggressively at inference-time if inference is the primary cost concern compared to an overtrained model served in 16-bit. Interestingly, the optimal data budget still increases as inference-time compute budget does, even though D doesn’t appear in the cost function. Further, the analysis reveals the insight that post-train precision also increases (logarithmically) with inference-time compute, for reasons we explore in the Appendix. Thanks for this suggestion!

---

> ### Author Response · Authors · 2024-11-20
> **Official Response by Authors, Part 2**
>
> - **Do the results generalize across hardware setups?**
>
> *Our results are hardware agnostic*. GPUs today broadly do not support matrix multiplication or computation in most of the precisions we study, so all our experiments simulate quantization instead of using literal low-precision types. So the statement that “The experiments are conducted on specific hardware setups that support low-precision computations, such as GPUs optimized for integer-type operations” is not quite true as stated, since we never actually use any low-precision capabilities of the hardware we train on.
>
> This is deliberate and standard in much work in quantization (eg. see [Wang et al, 2024]),and means that the trends we identify will generalize across hardware setups and remain relevant in the future. We do not assume anything about the hardware except for the fact that integer quantization (our focus) is implemented mathematically by some combination of rescaling and rounding, a standard convention [Jacob et al, 2018]. This desire to be hardware agnostic is also an important reason we don’t study or make any claims about performance metrics like latency or throughput, since these vary setup to setup. Even our optimality results just assume a particular functional form for loss and cost, and solve mathematically for the optima, without any hardware assumptions. **We’ve added discussion in Appendix D.3 clarifying this point.**
>
> - **Model size.**
>
> First and foremost, we want to be epistemically humble: certainly, the point of scaling laws is in some sense they are scale-invariant, but in practice many people believe that “more is different” so that models at trillion-parameter scale sometimes behave qualitatively differently than those at billion parameter scale. While the agreement between experiments at ~2B param scale and our fits at ~200m param scale suggest our scaling fits may indeed generalize, we welcome and look forward to follow-up work at larger model scale.
>
> A more general comment is that the contribution of the Chinchilla paper [Hoffmann et al, 2022], for instance, was not the literal prediction that the optimal amount of tokens is $D* = 20N$. Instead of using this verbatim, in practice training a real model for deployment involves fitting on one's own architecture/data, see eg. [Dubey et al, 2024]. Instead, one might argue that the contribution of the Chinchilla is the meta-observation that loss as a function of $N, D$ is predictable and the downstream insight that the resulting functional form implies that data and parameters should be scaled in equal proportion. In our setting, whether the compute-optimal training precision is literally 7-8 bits or some slightly different number based on a different architecture/data setup is rather beside the meta-point. The observation that loss as a function of precision can be predicted at all – and that this can have surprising implications of the kind we study – is an important meta-contribution, and this observation we believe will hold even at trillion parameter scale.
>
> **We believe our extensive set of new experiments and ablations in direct response to your questions make the manuscript significantly stronger. We’d appreciate it if you consider raising your score, or point us to further desiderata.** Thanks again for the detailed and thoughtful review!
>
>
> --
>
> **References.**
>
> Ma, Shuming, et al. "The era of 1-bit llms: All large language models are in 1.58 bits." arXiv preprint arXiv:2402.17764 (2024).
>
> Dubey, Abhimanyu, et al. "The llama 3 herd of models." arXiv preprint arXiv:2407.21783 (2024).
>
> Jacob, Benoit, et al. "Quantization and training of neural networks for efficient integer-arithmetic-only inference." Proceedings of the IEEE conference on computer vision and pattern recognition. 2018
>
> Hoffmann, Jordan, et al. "Training compute-optimal large language models." arXiv preprint arXiv:2203.15556 (2022).
>
> Wang, Hongyu, et al. "Bitnet: Scaling 1-bit transformers for large language models." arXiv preprint arXiv:2310.11453 (2023).

---

> > ### Comment · Reviewer_TyGC · 2024-11-21
> > **Acknowledgement**
> >
> > Thank you for the responses, your responses and additional experiments addresses my concerns and questions, so I'm raising the score.

---

### Official Review · Reviewer_jtGw · 2024-11-01

**Soundness:** 3
**Presentation:** 3
**Contribution:** 3
**Rating:** 8
**Confidence:** 3

**Summary:**

The paper studies the scaling law for precision, including exploring the #parameters, #tokens, pretraining precision, and inference precision.
The paper first introduces the background via (1) giving a decent introduction to quantization, (2) presenting the existing scaling laws on #parameters and #tokens, and (3) experimental setup.
Then the paper introduces the scaling laws for post-train quantization, and quantized training, sharing interesting findings.
Finally, a unified scaling law is introduced.

**Strengths:**

(1) The paper studies a meaningful topic, the scaling laws of precision, which is a new topic following the scaling law of data and parameters.

(2) The paper gives a good presentation. I especially appreciate the introduction to quantization. I'm not familiar with how quantization works in detail, so it helps a lot.

(3) The paper shows interesting findings in Sec. 3.1 Fig. 2: more pretraining tokens result in lower performance for post-train quantization with a high quantization rate.

(4) The paper shows interesting findings in Sec. 4.1 Fig. 3: KV cache is more sensitive to the change of precision when precision is low, but when precision is high, KV cache is more robust to the change of precision compared with weights and activations.

(5) The paper shows interesting findings in Sec. 4.3 Fig. 6: there would be cases where training in low precision leads to better evaluation loss.

(6) The paper generally shows that the proposed scaling law works well in the experimental setting of the paper.

**Weaknesses:**

(1) The paper uses the dataset Dolma for experiments. Though it's hard, it would be interesting to see how pretraining data affects this law.

(2) The paper uses the OLMo-style models for experiments. It would be great to give a general introduction to OLMo-style. Are they transformer-based model? While the abstract states the scaling law for language models, there would be other types of language models other than OLMo-style models, such as SSM.

**Questions:**

I respect the amount of experiments to support that the proposed scaling law works well. However, the counterintuitive findings are more attractive to me. The paper summarizes the findings in Fig 1. Could the author further explain the underline reasons/mechanism of such counterintuitive phenomenons?

---

> ### Author Response · Authors · 2024-11-20
> **Official Response by Authors**
>
> Thank you for the thoughtful and kind review!
>
> - We ablate the dataset to use C4 (albeit at small scale) in Appendix A, finding similar trends.
> - The models we train are indeed standard causal Transformer-based language models. We’ve now mentioned their architecture in  Appendix A, and exploring whether these trends are universal across classes of models (Transformers vs SSMs, etc) is an exciting line of future work!
>
> Re: the counterintuitive findings. For post-train quantization, we include a discussion on what could be mechanistically causing it in Appendix H – our working intuition is that as models are trained on more data, they store more information in their weights in a more compressed manner, so that noise of a fixed variance to perturb those weights will have a relatively more damaging effect. And about quantized training, our intuition is that we can think of quantization as applying noise of a variance that decreases quickly with precision on the forward pass. So the precision-parameter tradeoff at the center of our quantized training section is really just a matter of asking: do we expect a 1b model with noise variance 1 on its forward pass to do better than a 200m model with no noise? It does not seem obvious beforehand what the answer to such a question is, which is why a scaling law is needed. This assumes that quantization effects on the forward pass can reasonably be thought of intuitively as noise, which seems to be justified by the fact that our functional forms fit the data well. Thanks for the good questions!
>
> We’ve also added a number of things to the manuscript that we think substantially strengthen it, and uploaded a new version with new Appendix sections in blue:
> - We’ve **added preliminary experiments studying quantized training with FP instead of INT, involving over 100 new pretraining runs.** Going beyond our functional form, we examine how loss scales with FP bits under non-standard mantissa vs exponent bit allocations. See Appendix N.
> - We’ve **added a new inference cost model studying the optimal choice of overtraining vs post-train quantization for workloads where inference is the dominant cost.** This complements our training cost analysis to now provide insights for practitioners who care mostly about inference. It can be found in Appendix E.3.
> - We’ve **added experiments in Appendix I testing the effect of per-tensor vs per-channel quantization** on scaling behavior with weight vs activation precision, finding that while granularity can explain part of the difference between weights and activations, activations are indeed inherently more sensitive to quantization at the same quantization granularity.
> - We’ve added minor improvements throughout the manuscript: reformatting some plots, changing our validation shard, rewriting some paragraphs for clarity, and including more implementation details.
> - We’ve **added ablations for post-train-quantization**: we reproduce our degradation results using AWQ, a modern post-train quantization method, as well as when we train with a linear learning rate schedule instead of cosine. The behavior persists under both ablations. See new Appendix F and G.
>
> We believe these additions make the manuscript much more complete and comprehensive. If there’s any experiment that would further improve your assessment of our work, feel free to let us know. Many thanks for the review!

---

> > ### Comment · Reviewer_jtGw · 2024-11-21
> > **Ack**
> >
> > Thank you for your response. I'll keep my score and vote to accept the paper.

---

### Official Review · Reviewer_Xi3X · 2024-11-02

**Soundness:** 3
**Presentation:** 3
**Contribution:** 3
**Rating:** 8
**Confidence:** 3

**Summary:**

This manuscript provides a thorough investigation into the impact of bit precision on inference performance and introduces a scaling law that correlates performance with precision. The paper is commendable for its extensive experimental validation. The study addresses a significant problem in the field of deep learning optimizations and offers practical insights for efficient model deployment. The manuscript is well-structured and the arguments are clearly presented

**Strengths:**

Strengths:

1. The paper tackles an important issue with the introduction of a bit precision scaling law. While this topic has been explored before, the theoretical scaling law presented in this work offers valuable guidance for the efficient deployment of models in real-world applications. The implications of this work could be transformative for the field.

2. The authors have provided a wealth of experimental results that not only validate the existing scaling laws across different model sizes but also demonstrate the generalizability of previously unseen scenarios. This thorough experimental section strengthens the paper's contributions and is persuasive.

3. The manuscript is particularly strong in its methodological rigor, with a clear articulation of the scaling laws and their implications for precision in deep learning models.

**Weaknesses:**

no clear weakness.

**Questions:**

potential typos:

1. row303: P_a and =P_{kv} as well

---

> ### Author Response · Authors · 2024-11-20
> **Official Response by Authors**
>
> Thank you for the review! The typo is fixed.
>
> We’ve made a number of significant improvements to the manuscript and uploaded a new version with new Appendix Sections in blue to easily find them:
> - We’ve **added preliminary experiments studying quantized training with FP instead of INT, involving over 100 new pretraining runs.** Going beyond our functional form, we examine how loss scales with FP bits under non-standard mantissa vs exponent bit allocations. See Appendix N.
> - We’ve **added a new inference cost model studying the optimal choice of overtraining vs post-train quantization for workloads where inference is the dominant cost.** This complements our training cost analysis to now provide insights for practitioners who care mostly about inference. It can be found in Appendix E.3.
> - We’ve **added experiments in Appendix I testing the effect of per-tensor vs per-channel quantization** on scaling behavior with weight vs activation precision, finding that while granularity can explain part of the difference between weights and activations, activations are indeed inherently more sensitive to quantization at the same quantization granularity.
> - We’ve added minor improvements throughout the manuscript: reformatting some plots, changing our validation shard, rewriting some paragraphs for clarity, and including more implementation details.
> - We’ve **added ablations for post-train-quantization**: we reproduce our degradation results using AWQ, a modern post-train quantization method, as well as when we train with a linear learning rate schedule instead of cosine. The behavior persists under both ablations. See new Appendix F and G.
>
> We believe these additions substantially strengthen the manuscript; if there’s any experiment that you think would further improve your assessment of our work, please let us know. Many thanks for the positive and thoughtful review.

---

### Meta-Review · Area_Chair_YPXb · 2024-12-20

**Metareview:**

This paper investigates the scaling laws for precision, focusing on factors such as the number of parameters, tokens, pretraining precision, and inference precision. It provides a comprehensive background of an overview of quantization, a review of existing scaling laws for parameters and tokens, and a description of the experimental setup. The study then introduces scaling laws for post-training quantization and quantized training, highlighting notable findings. Lastly, it proposes a unified scaling law that integrates these insights.

The reviewers concur that this paper makes a significant contribution to the field. It excels in methodological rigor, offering a clear and well-articulated analysis of scaling laws and their implications for precision in deep learning models. The introduction of a bit-precision scaling law addresses an important issue, providing a theoretical framework that builds upon prior work. This novel scaling law offers practical insights for the efficient deployment of models in real-world scenarios, with the potential to drive transformative advancements in the field. The experimental results are convincing which not only validate the existing scaling laws across different model sizes but also demonstrate the generalizability of previously unseen scenarios.

There are also several weaknesses pointed out by the reviewers, which have been addressed during the discussion phase. Therefore, I would recommend acceptance of this work. I encourage the authors to incorporate the discussions in their next version.

**Additional Comments On Reviewer Discussion:**

There are several weaknesses pointed out by the reviewers, which have been addressed during the discussion phase.

---

### Decision · Program_Chairs · 2025-01-22

Accept (Oral)